# Visual and motor signatures of locomotion dynamically shape a population code for feature detection in *Drosophila*

Maxwell H Turner, Avery Krieger, Michelle M Pang, Thomas R Clandinin*

Department of Neurobiology, Stanford University, Stanford, United States

**Abstract** Natural vision is dynamic: as an animal moves, its visual input changes dramatically. How can the visual system reliably extract local features from an input dominated by self-generated signals? In *Drosophila*, diverse local visual features are represented by a group of projection neurons with distinct tuning properties. Here, we describe a connectome-based volumetric imaging strategy to measure visually evoked neural activity across this population. We show that local visual features are jointly represented across the population, and a shared gain factor improves trial-to-trial coding fidelity. A subset of these neurons, tuned to small objects, is modulated by two independent signals associated with self-movement, a motor-related signal, and a visual motion signal associated with rotation of the animal. These two inputs adjust the sensitivity of these feature detectors across the locomotor cycle, selectively reducing their gain during saccades and restoring it during intersaccadic intervals. This work reveals a strategy for reliable feature detection during locomotion.

## Editor's evaluation

This manuscript investigates how the fly visual system can encode specific features in the presence of self-generated motion. Using volumetric imaging, it explores the encoding of visual features in population activity in the *Drosophila* visual glomeruli – a set of visual "feature detectors". Through an elegant combination of neural imaging, visual stimulus manipulations, and behavioral analysis, it demonstrates that two different mechanisms, one based on motor signals and one based on visual input, serve to suppress local features during movements that would corrupt these features. The results of this study open up new directions to determine how motor and visual signals are integrated into visual processing at the level of neural circuits.

**\*For correspondence:**
trc@stanford.edu

**Competing interest:** The authors declare that no competing interests exist.

## Introduction

Sighted animals frequently move their bodies, heads, and eyes to achieve their behavioral goals and to actively sample the environment. As a result, the image on the retina is frequently subject to self-generated motion. This presents a challenge for the visual system, as visual circuitry must extract and represent specific features of the external visual scene in a rapidly changing context where the dominant sources of visual changes on the retina may be self-generated. While this problem has been well studied in the context of motion estimation (*Borst et al., 2010*; *Britten, 2008*), the broader question of how visual neurons might extract local features of the scene under naturalistic viewing conditions is relatively poorly understood. How do visual neurons selectively encode local features of interest under these dynamic conditions?

DOI: https://doi.org/10.7554/eLife.82587

Local feature detection during self-motion presents unique challenges. For detecting widefield motion, or large static features of the scene like oriented edges and landmarks, the visual scene is intrinsically redundant, as many neurons distributed across the visual field can encode information that is relevant to the feature of interest even as the scene moves. Conversely, local features like prey, conspecifics, or approaching predators engage only a small part of the visual field, dramatically reducing the redundancy of the visual input. In addition, neurons that selectively respond to small features could also be activated by high spatial frequency content in the broader scene, potentially corrupting their responses under naturalistic viewing conditions. Neurons that respond selectively to local visual features have been described in many species, including flies, amphibians, rodents, and primates (*Keleş and Frye, 2017*; *Kerschensteiner, 2022*; *Klapoetke et al., 2022*; *Lettvin et al., 1959*; *Pasupathy and Connor, 2001*; *Piscopo et al., 2013*). However, these studies have typically been conducted either in non-behaving animals, or under conditions of visual fixation. Here, we explore the neural mechanisms by which local feature detection is made robust to the visual inputs and behavioral signals associated with natural vision.

Strategies for reliable visual feature detection during self-motion fall into one of at least three categories. First, behavioral strategies can help mitigate the impact of self-motion on visual feature encoding by changing the nature of the neural encoding task at hand. For example, compensatory movements of the eyes, head, or body can stabilize the image on the retina during self-motion (*Angelaki and Hess, 2005*; *Hardcastle and Krapp, 2016*; *Land, 1999*; *Walls, 1962*), and saccadic movement dynamics compress the fraction of time during which large self-generated motion signals corrupt retinal input (*Martinez-Conde et al., 2013*; *Van Der Linde et al., 2009*; *Wurtz, 2018*; *Cruz et al., 2021*; *Geurten et al., 2014*; *Collett and Land, 1975b*). In other cases, behavior is shaped by the demands of a specific visual task. For example, dragonflies and other predatory insects often approach prey from below, increasing the likelihood that a target will be seen against a background of the low contrast sky (*Nordström and O'Carroll, 2009*), and male hoverflies hover in place while monitoring for conspecific territorial trespassers (*Collett and Land, 1975a*), ensuring that self-generated motion signals are low during a demanding visual discrimination task.

Second, neural mechanisms can exploit the fact that self-generated motion produces characteristic sensory inputs. For example, visual surrounds can be tuned to the global motion signals characteristic of self-motion, allowing for self-motion signals to be subtracted from excitatory center signals that code for a feature of interest (*Aptekar et al., 2015*; *Baccus et al., 2008*; *Olveczky et al., 2003*; *Egelhaaf, 1985*; *Collett, 1971*). However, in some flying insects, target-detecting neurons are tightly tuned for very small visual targets (*Nordström and O'Carroll, 2006*), even in the context of moving, cluttered backgrounds (*Nordström et al., 2006*; *Wiederman and O'Carroll, 2011*), suggesting that multiple levels of spatial inhibition can work together to shape feature selectivity (*Bolzon et al., 2009*), and that robust feature detection need not rely on relative motion cues (*Nordström, 2012*; *Nordström and O'Carroll, 2009*; *Wiederman et al., 2008*).

The third strategy for reliable vision during self-motion uses signals related to the animals' motor commands or behavioral states to modulate neural response gain. For example, the motor commands that initiate primate saccades produce efference copy signals that are associated with neural gain changes and a perceptual decrease in sensitivity called saccadic suppression (*Binda and Morrone, 2018*; *Bremmer et al., 2009*; *Wurtz, 2018*). In flies, efference copy signals can cancel expected motion in widefield motion-sensitive neurons during flight (*Fenk et al., 2021*; *Kim et al., 2015*; *Kim et al., 2017*), but can also provide independent information about intended movements (*Fujiwara et al., 2017*; *Fujiwara et al., 2022*; *Cruz et al., 2021*). In this way, neural response gain is modulated so that motion-sensitive neurons encode unexpected deviations in motion signals after accounting for behavior.

Previous studies have each examined these respective strategies in the context of single-cell types. However, how do these varied strategies work together across a population of disparately tuned visual neurons? We explore this issue using populations of visual projection neurons (VPNs) in *Drosophila*. VPNs are situated at a critical computational and anatomical bottleneck through which highly processed visual information moves from the optic lobes to the central brain. A subset of VPNs, the Lobula Columnar (LC) and Lobula Plate Lobula Columnar (LPLC) cells (*Fischbach and Dittrich, 1989*; *Otsuna and Ito, 2006*; *Wu et al., 2016*) make up a large fraction of all VPN types, thus accounting for a substantial portion of the visual information available to guide behavior. These cell types encode

distinct local visual features with behavioral relevance, including looming objects (*Ache et al., 2019*; *Klapoetke et al., 2017*) and small moving objects (*Keleş and Frye, 2017*; *Ribeiro et al., 2018*) (for a recent survey of VPN visual tuning, see *Klapoetke et al., 2022*), and project to small, distinct regions in the central brain called optic glomeruli (*Wu et al., 2016*; *Panser et al., 2016*). Previous work has also implicated some types of LCs in figure-ground discrimination, that is, the ability to detect an object moving independently of a global background motion signal (*Aptekar et al., 2015*). Each optic glomerulus receives input from all of the individual cells belonging to a single-cell type, resulting in a functional map in the central brain (*Klapoetke et al., 2022*). Moreover, both stimulation and silencing experiments argue that at least some VPN classes strongly modulate specific visually guided behaviors (*Hindmarsh Sten et al., 2021*; *Tanaka and Clark, 2020*; *Tanaka and Clark, 2022*). Finally, the visual tuning of VPN types is heterogeneous across the population, allowing us to explore how strategies for reliable visual encoding during self-motion vary across differently tuned populations.

To explore how local visual features are represented across populations of VPNs, we developed a new method to register functional imaging data to the fruit fly connectome, allowing us to measure neural responses across many optic glomeruli simultaneously. We show that this method allowed for reliable and repeatable measurement of VPN responses. This population imaging method allowed us to measure the covariance of optic glomerulus population responses to visual stimuli. This analysis revealed strongly correlated trial-to-trial variability across glomeruli, which improves stimulus encoding fidelity. Importantly, this could not have been inferred from non-simultaneous measurements. We next demonstrate that walking behavior selectively suppressed responses of small object detecting glomeruli, leaving responses to looming objects unchanged. We then focus on body rotations as an example of self-motion that introduces large, uniform displacements in visual input during behavior to show that visual stimuli characteristic of rotational self-motion, including those produced by locomotor saccades, also suppressed VPN responses to small objects. Finally, we show that these two forms of gain control—visual and motor-associated—can be independently recruited and reinforce one another when both are active. Taken together, these results reveal that both visual and motor cues associated with self-motion can tune local feature detecting VPNs, adjusting their sensitivity to match the dynamics of natural walking behavior. This suggests a strategy for resolving the ambiguities associated with detecting external object motion in a scene dominated by self-generated visual motion.

## Results
### Visual rotation complicates local feature detection

To build intuition about how self-generated motion might impact local feature selectivity, we designed a task inspired by VPN selectivity to small, moving objects (*Keleş and Frye, 2017*; *Klapoetke et al., 2022*), and by target discrimination tasks performed by other flying insects (*Egelhaaf, 1985*; *Nordström and O'Carroll, 2006*). For this analysis, we focused on the impact of rotational self-motion, because it is a prominent component of self-generated optic flow during movement that causes large movement signals that are uniform across the visual field. In this detection task, a 15° dark patch moved on top of a grayscale natural image background, through a receptive field whose size was typical of small object detecting LCs (*Figure 1A*). When the natural image background was static, as would be the case if a stationary fly were observing an external moving object in a rich visual environment, detecting the moving patch is trivial given the change in local luminance and/or spatial contrast as the patch traverses the receptive field (*Figure 1B*). How is this detection task impacted by rotational self-motion? We simulated self-generated rotational motion by moving the background image at a single, constant velocity (*Figure 1C*). This background motion caused large fluctuations in local luminance and spatial contrast, reflecting the heterogeneous spatial structure of the scene (*Figure 1D*, red traces). These fluctuations were often larger than the changes induced by the moving patch alone (e.g., compare *Figure 1B* to *Figure 1D*). Moreover, with an independently moving patch added to the foreground, the change in local luminance or contrast was negligible for this example image (*Figure 1D*, blue traces), making discrimination between these two conditions very difficult.

We quantified discriminability, d', between traces where only the background image moved and traces where the small patch moved on top of the moving background, using either local luminance signals (*Figure 1E*, left) or local contrast signals (*Figure 1E*, right). This metric captures the difference between the mean responses to 'spot present' versus 'spot absent' normalized by the standard

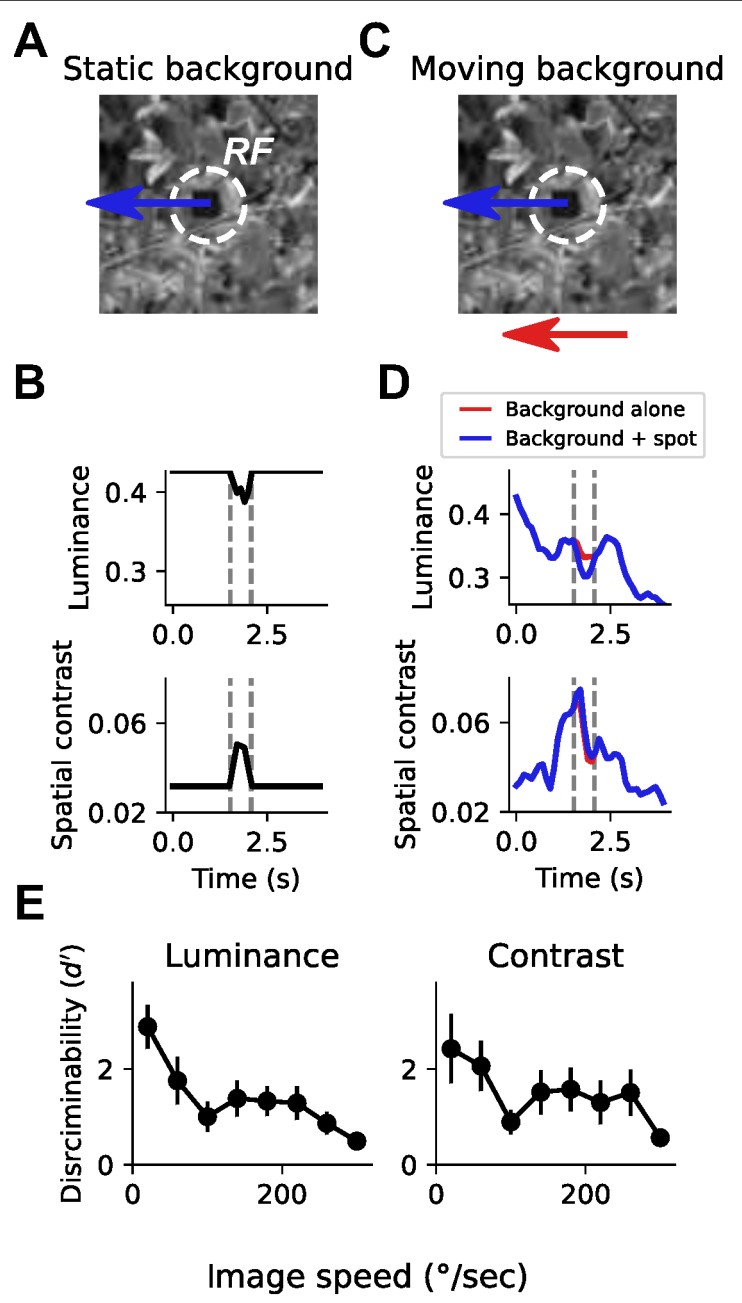

**Figure 1.** Small spot detection is unreliable during self-generated rotation. (**A**) Schematic illustrating the stimulus and detection task. A small dark spot on top of a grayscale natural image moved through a visual receptive field (white dashed circle). (**B**) When the background image was stationary, small spot detection was trivial using either local luminance (top) or spatial contrast (bottom) cues. Vertical dashed lines indicate the window of time that the spot passes through the receptive field. (**C**) To mimic an object detection task during self-generated rotation, we moved the background image independently of the spot with a variable speed. (**D**) Movement of the background image alone (red trace) caused dramatic fluctuations in luminance and contrast within the receptive field. The addition of the small moving spot (blue trace) caused relatively small changes in the luminance or contrast signal, which depends on the spatial structure of the image. (**E**) Discriminability of the spot based on luminance (left) or contrast (right) during the time period when the spot passed through the receptive field, as a function of background image speed. Points indicate mean ± S.E.M. across a collection of 20 grayscale natural images. Even for slow background speeds, detection was corrupted, and the discriminability of the spot decreased further as the background speed increased.

deviation of the response traces (see Materials and methods). d′ reflects the z-scored difference between the responses to these two conditions, meaning that a d′ of 0 corresponds to chance under an ideal observer model. With a static or absent background, the discriminability of the patch is perfect. Across a collection of 20 natural images (*van Hateren and van der Schaaf, 1998*), moving at velocities between 20°/s and 320°/s, small object detection was corrupted even for small amounts of background motion, and discriminability decreased further as background motion increased (*Figure 1E*). These observations suggest that as self-motion signals increase, neurons that respond selectively to local features like small moving objects might increase their response thresholds in order to avoid relaying false positive signals.

## A connectome-based alignment method to measure population activity across optic glomeruli

To efficiently characterize the responses of individual VPNs to many visual stimuli, and to relate the gain of multiple VPNs with one another and to animal behavior, we needed to measure responses across different VPN types simultaneously. Presently, specific driver lines exist to target single VPN types in a single experiment (*Wu et al., 2016*), but no approach exists to measure across many VPN types simultaneously. To develop such a population recording approach, we exploited the fact that optic glomeruli are physically non-overlapping (*Figure 2A*). Each optic glomerulus receives dominant input from one type of LC or LPLC cell (with one known exception being LPLC4/LC22 *Wu et al., 2016*, not included in this study). At the same time, the fly brain is highly stereotyped, meaning that by aligning functional imaging data to the *Drosophila* connectome (*Scheffer et al., 2020*), we could use the positions of VPN presynaptic active zones (T-bars) to identify voxels that correspond to specific glomeruli.

We selected the optic glomeruli in the Posterior Ventrolateral Protocerebrum (PVLP) and Posterior Lateral Protocerebrum (PLP) for imaging (*Figure 2A*), because this region of the brain contains the majority of known optic glomeruli in a confined volume. We imaged the left PVLP/PLP using a two-photon resonant scanning microscope, which allowed for sampling of the volume of interest at ~7 Hz (*Figure 2B*, see Materials and methods). As previous work had demonstrated that individual VPN cells respond to visual stimuli with monophasic calcium responses that span several hundred milliseconds (as measured using GCaMP6f *Klapoetke et al., 2022*), this volume rate provides dense temporal sampling of each VPN type.

Optic glomeruli contain neurites from many neuron types, including the presynaptic terminals of their dominant VPN input, but also postsynaptic targets of those cells as well as other local interneurons. We used a two-pronged approach to bias measured calcium signals toward those selective to presynaptic terminals of VPNs. First, we developed a GCaMP6f variant that preferentially localizes to presynaptic terminals (syt1GCaMP6f). This construct showed much brighter GCaMP6f fluorescence in axon terminals in the optic glomerulus compared to dendrites in the lobula (*Figure 2C*). Second, as almost every LC and LPLC neuron is cholinergic, we specifically targeted cholinergic neurons using a ChAT-T2A knock-in Gal4 driver line (*Deng et al., 2019*). Using this driver line, we expressed both syt1GCaMP6f as well as myr::tdTomato, a plasma-membrane bound red structural indicator that was used for motion correction and alignment (*Figure 2D*).

To extract glomerulus responses from our in vivo imaging volumes, we used techniques similar to other recent imaging alignment studies in the *Drosophila* brain (*Brezovec et al., 2022*; *Mann et al., 2017*; *Pacheco et al., 2021*; *Turner et al., 2021*). First, we generated a 'mean brain' volume by iteratively aligning and averaging a collection of high-resolution, in vivo anatomical scans of the volume of interest (*Figure 2E*, n=11 flies). Next, we used the syt1GCaMP6f channel of the mean brain to align to the JRC2018 template brain (*Figure 2F*; *Bogovic et al., 2020*). Finally, we generated a glomerulus map using locations of the presynaptic T-bars belonging to LC and LPLC neurons, which we extracted from the hemibrain connectome (*Scheffer et al., 2020*), and aligned it to the JRC2018 template brain. Using the mean brain and mean brain-template alignment, we could consistently align individual volumes to the mean brain and to the glomerulus map (*Figure 2G and H*). This method, which we refer to as pan-glomerulus imaging, allowed us to assign voxels in a single fly's in vivo volume to a specific optic glomerulus. In this paper, we focus on 13 glomeruli (*Figure 2F*, *Figure 2—figure supplement 1*).

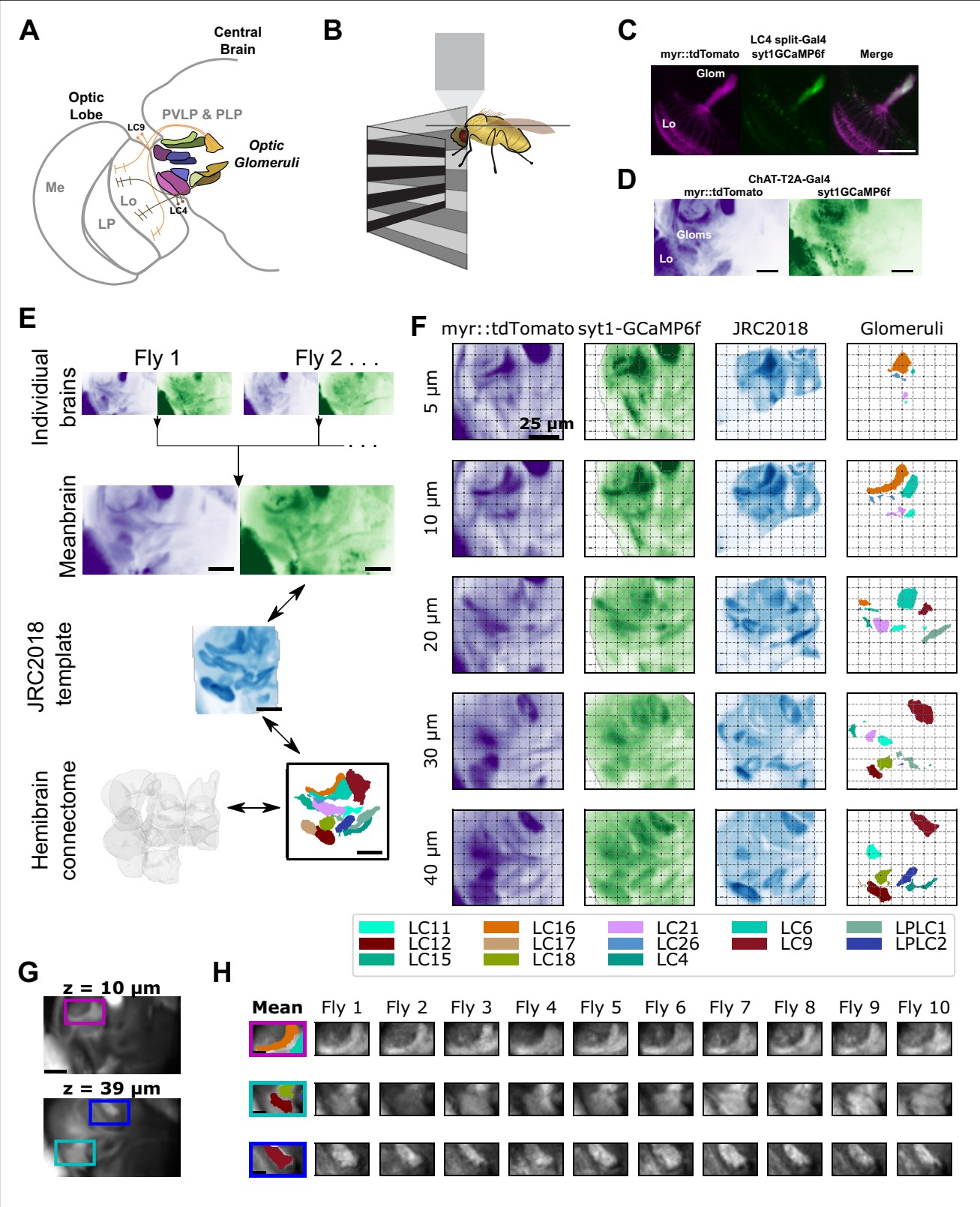

**Figure 2.** A method to extract optic glomerulus population responses from bulk-labeled neuropil. (**A**) Schematic of the left half of the brain showing optic lobe and the optic glomeruli of the central brain, which receive inputs from distinct visual projection neurons. Me: Medulla, LP: Lobula Plate, Lo: Lobula, PVLP: Posterior Ventrolateral Protocerebrum, PLP: Posterior Lateral Protocerebrum. (**B**) Schematic of imaging and stimulation setup. (**C**) LC4 neurons expressing plasma membrane-bound myr::tdTomato (magenta) and presynaptically localized syt1GCaMP6f (green), which is enriched in axons

*Figure 2 continued on next page*

*Figure 2 continued*

in the optic glomerulus. (**D**) For pan-glomerulus imaging, cholinergic neurons express myr::tdTomato (purple) and syt1GCaMP6f (green). The optic glomeruli in the PVLP/PLP can be seen. (**E**) Pipeline for generating the mean brain from in vivo, high-resolution anatomical scans and aligning this mean brain to the JRC2018 template brain. Using this bridging registration, neuron and presynaptic site locations from the hemibrain connectome can be transformed into the mean brain space, allowing in vivo voxels to be assigned to distinct, non-overlapping optic glomeruli. (**F**) Montage showing z planes (rows) of the registered brain space for the mean brain myr::tdTomato (purple) and syt1GCaMP6f (green) channels (first and second columns, respectively), JRC2018 template brain (third column) and optic glomeruli map (fourth column). (**G**) Mean brain images at indicated z levels showing distinct glomerulus locations of interest. (**H**) For the locations of interest in (**G**), the optic glomerulus map is overlaid on the mean brain (first column) and alignment is shown for each of the 10 individual flies (remaining columns). For all images, scale bar is 25 μm.

The online version of this article includes the following figure supplement(s) for figure 2:

**Figure supplement 1.** Number of voxels in each LC/LPLC glomerulus.

To test whether pan-glomerulus imaging reliably captured visually driven calcium responses across glomeruli, we presented a suite of synthetic stimuli meant to explore VPN feature detection (*Keleş and Frye, 2017*; *Klapoetke et al., 2017*; *Klapoetke et al., 2022*; *Wu et al., 2016*). Our stimulus suite therefore consisted of small, moving spots, static flicker, looming spots, moving bars, and other stimuli (see Materials and methods). *Figure 3A* shows mean glomerulus responses across animals to these stimuli. As expected, the visual tuning measured in one glomerulus in one fly was very similar to tuning seen in corresponding glomeruli measured in other animals (*Figure 3B and C*). To determine whether our pan-glomerulus imaging method accurately captured the visual tuning of the VPN that provides the major input to that glomerulus, we used cell-type-specific split-Gal4 driver lines for select VPN types (LC18, LC9, and LC4), chosen because together they span the anatomical volume of interest, and presented the same stimulus suite (*Wu et al., 2016*). We then compared these targeted recordings to those previously measured in the corresponding glomeruli, using our population imaging approach. For each of these VPN/glomerulus pairs, the responses and visual tuning looked qualitatively similar (*Figure 3D*) and were highly correlated (*Figure 3E*). Taken together, these results show that pan-glomerulus imaging reliably measures visually driven responses across a population of optic glomeruli, and that these visual responses are dominated by VPN signals.

At a high level, this initial suite of stimuli revealed that optic glomeruli show broad, overlapping tuning (*Figure 3A*) in line with previous observations using cell-type-specific driver lines (*Klapoetke et al., 2022*). To conveniently organize the results presented in subsequent analyses, we applied a hierarchical clustering approach to identify functional groupings of VPN types based on their responses to our synthetic stimulus suite. Group 1 was characterized by LCs that responded to moving spots 5° in diameter (the smallest stimuli presented here), and showed relatively weak responses to loom and vertical bars. Group 2 contained glomeruli that were not sensitive to very small objects and showed strong loom responses. Group 3 contained glomeruli that were typically only weakly driven by any of these stimuli but responded to looming stimuli. Finally, group 4 glomeruli had large responses to vertical bars and medium and large moving spots as well as some loom sensitivity.

## Population activity is modulated by a dominant gain factor which impacts stimulus coding fidelity

Previous characterization of VPNs relied on targeting each individual cell class using cell-type-specific driver lines (*Klapoetke et al., 2022*; *Wu et al., 2016*). This allows for the measurement of neural response mean and variance, but not the covariance among different VPNs, which requires simultaneous measurement. Trial-by-trial covariance can have a dramatic impact on stimulus encoding (*Averbeck and Lee, 2006*; *Averbeck et al., 2006*; *Romo et al., 2003*; *Zylberberg et al., 2016*), and can shed light on the circuit mechanisms that govern sensory computation (*Ala-Laurila et al., 2011*; *Rabinowitz et al., 2015*). To examine the covariance structure of optic glomerulus responses, we presented a subset of the synthetic stimuli (*Figure 3*), and collected 30 trials for each stimulus. We observed significant trial-to-trial variability. Indeed, on some presentations of a stimulus which, on average, drives a strong response, many glomeruli failed to respond at all. Moreover, this large modulation in response gain was shared across many glomeruli on a trial-by-trial basis (*Figure 4A and B*). When we averaged the trial-to-trial correlations across flies, we observed strong, positive pairwise correlations across the glomerulus population (*Figure 4C*), and across stimuli (*Figure 4—figure supplement 1*). Because we also collected myr::tdTomato fluorescence through the red channel, we

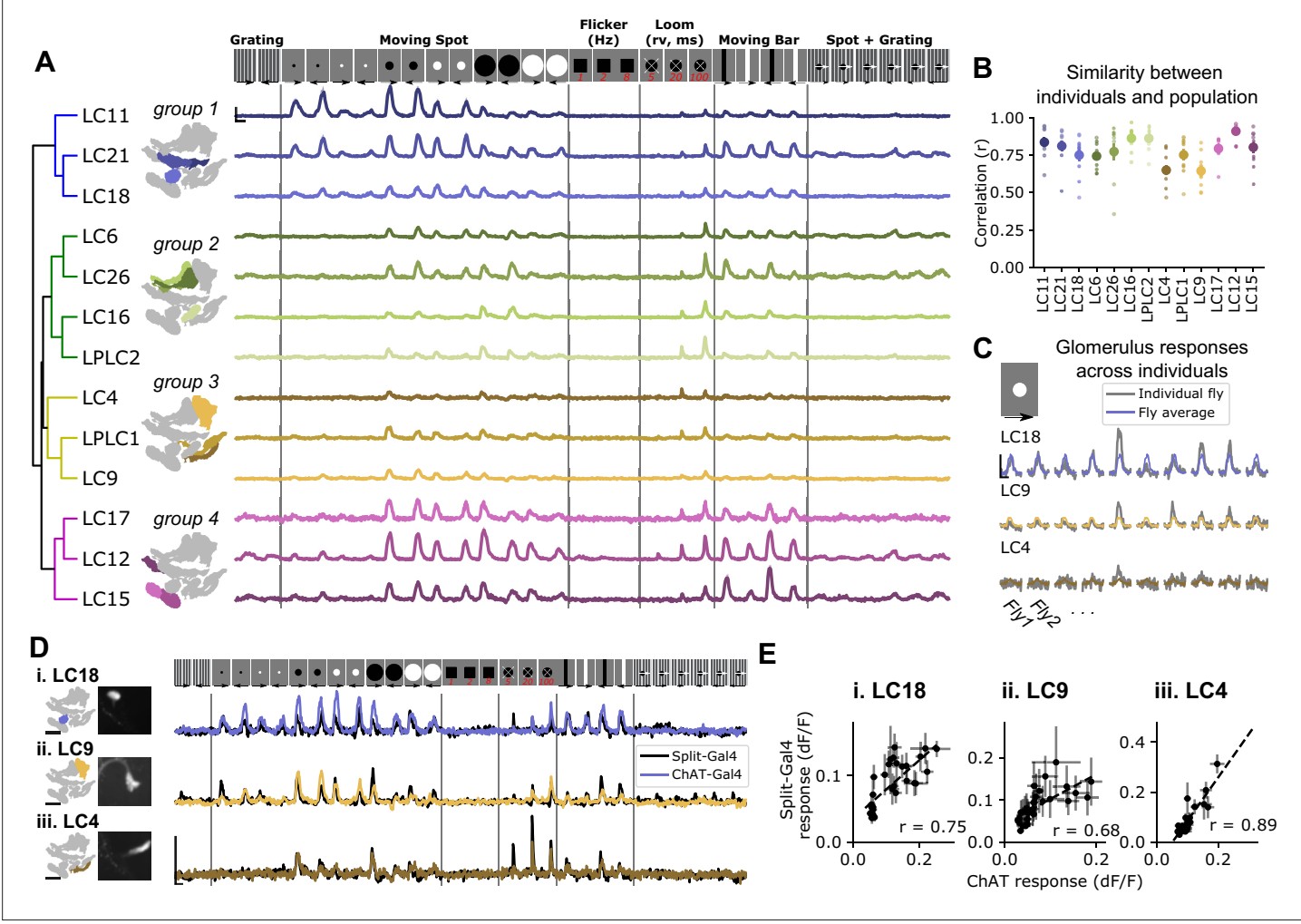

**Figure 3.** Pan-glomerulus imaging reliably measures optic glomerulus responses dominated by visual projection neurons. (**A**) Responses of thirteen optic glomeruli to a panel of synthetic visual stimuli (top, see Materials and methods). Responses are shown according to the indicated stimulus order, but stimuli were presented in randomly interleaved trial order. Shown are mean glomerulus response traces across 10 flies, shading indicates S.E.M. We hierarchically clustered mean glomerulus responses to yield four functional groups of glomeruli. (**B**) Visual tuning can be reliably estimated in single flies using pan-glomerulus imaging. For each glomerulus, we computed the correlation between each individual fly tuning and the mean tuning (excluding that fly). Large dots and bars indicate mean ± S.E.M., and small dots correspond to individual flies. (**C**) Example responses of three glomeruli in individual flies to a 15° bright moving spot. For each panel, the colored trace is the across-fly average response and the gray trace is the individual fly response. (**D**) Comparison of glomerulus tuning to the LC neurons that dominate their input, for three glomerulus/LC pairs. Left: glomerulus map and example image of the LC axons. Right: syt1GCaMP6f responses to the stimulus panel above. Colored traces show tuning measured using pan-glomerulus imaging procedure and black traces show split-Gal4 LC responses. (**E**) For the three LC types in (**D**), the tuning of LC axons is highly correlated with the tuning of corresponding optic glomeruli (pan-glomerulus imaging: n=10 flies; Split-Gal4 imaging: n=5, 6, and 4 flies for LC18, LC9, and LC4, respectively). For all calcium traces, Scale bar is 2 s and 25% dF/F. r is the Pearson correlation coefficient.

could use this structural signal to assess whether the trial covariance we observed in syt1GCaMP6f responses was due to other factors not associated with neural responses, like brain motion that was not removed during motion correction, or other imaging factors. As expected, myr::tdTomato signals showed very little modulation across trials (*Figure 4A*, gray traces), and as a result the trial to trial covariance was weaker and showed a qualitatively different structure than the covariance in syt1G-CaMP6f signals (*Figure 4—figure supplement 2*). This indicates that trial covariance in syt1GCaMP6f signals was dominated by visually driven responses.

This large response variance suggests a challenge for downstream circuits integrating information across optic glomeruli: how can a visual feature be reliably decoded when response strength shows

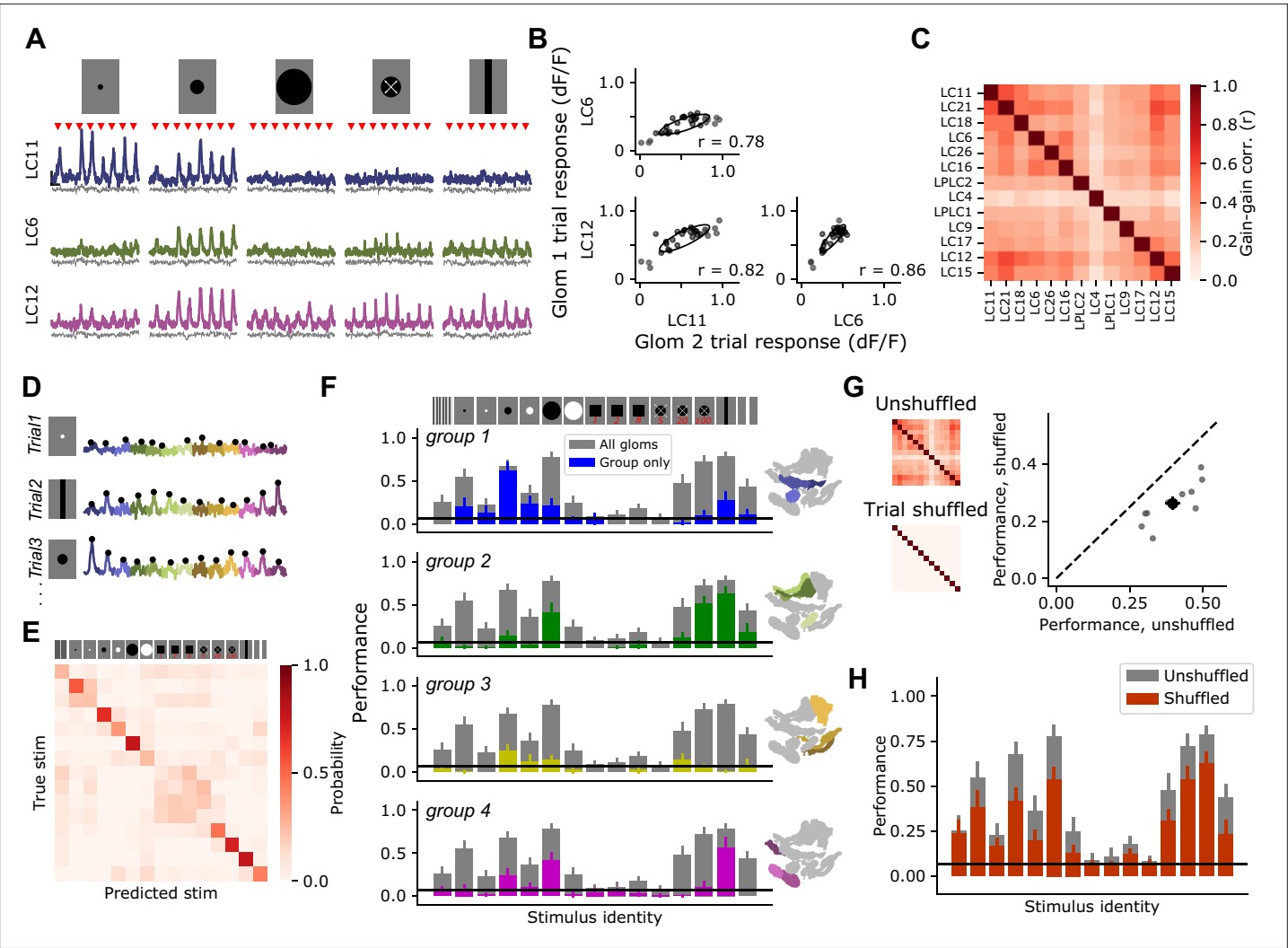

**Figure 4.** Glomerulus responses are modulated by a shared gain factor that impacts stimulus encoding fidelity. (**A**) For a reduced stimulus set (images above), we presented many trials in a randomly interleaved order. For display, we have grouped responses by stimulus identity. Red marks indicate stimulus presentation times. Example single-trial responses for representative glomeruli show large trial-to-trial variability. Scale bar is 4 s and 25% dF/F. Gray traces below show simultaneous glomerulus signals from the red (myr::tdTomato) channel showing that fluorescence changes are due to visually driven responses, not motion artifacts or other imaging factors. (**B**) For the example glomeruli in (**A**), plotting one glomerulus response amplitude against another for a given stimulus (here, a 15° moving spot), reveals high correlated variability. Each point is a trial. Ellipses show 2D Gaussian fit derived from the trial covariance matrix. r is the Pearson correlation coefficient. (**C**) Trial-to-trial variability is strongly correlated across different glomeruli. Heatmap shows the average correlation matrix across all stimuli and flies (n=17 flies). (**D**) Single-trial responses of 13 optic glomeruli, concatenated together for each trial. Stimulus identity is indicated to the left. Black dots indicate the peak responses of each glomerulus on each trial, which will be used for stimulus decoding. The response amplitudes for all 13 glomeruli were used to train a multinomial logistic regression model to predict stimulus identity (see Materials and methods). (**E**) Confusion matrix, with rows and columns corresponding to stimulus identity above. (**F**) We used held-out data to test the ability of the decoding model to predict the stimulus identity given the single-trial response amplitudes of different groupings of glomeruli. In each barchart, gray bars correspond to a model with access to all 13 glomerulus responses for each trial. Colored bars show mean ± S.E.M. performance of a model with access to only the indicated functional group of glomeruli. (**G**) Trial shuffling population responses remove pairwise correlations among glomeruli (insets to the left show correlation matrices before and after trial shuffling). Right: Decoding model performance, averaged across all stimuli, for each fly (n=11 flies). Large marker shows mean ± S.E.M. (**H**) Performance of the decoding model suffers across all stimulus classes when trial-to-trial correlations are removed.

The online version of this article includes the following figure supplement(s) for figure 4:

**Figure supplement 1.** Trial correlation matrix for each stimulus class.

**Figure supplement 2.** Trial covariance matrix for both myr::tdTomato and sytGCaMP6f fluorescence channels.

**Figure supplement 3.** Decoding stimulus identity for a reduced subset of behaviorally discriminable stimulus classes.

such large variability from trial to trial? To explore this issue, we implemented a multinomial logistic regression decoder to predict the identity of a stimulus given single-trial population responses. Since the animal does not have a priori information about when or where a local visual feature might appear, we did not want the model to be able to use different stimulus dynamics to trivially learn the decoding task based on response timing. Therefore, we trained the model using only the peak response amplitude from each glomerulus on each trial (*Figure 4D*), and tested the ability of the model to predict stimulus identity on held-out trials. This decoding model performed with an overall accuracy rate of around 40%, on average (compared to a chance performance of 7%), and performance for some stimulus classes was considerably higher (*Figure 4E*). For example, for dark moving spots with diameter 5°, 15°, and 50°, performance was 55%, 67%, and 78%, respectively. For a slowly looming spot, performance was 72%. This high performance was surprising given that the model only had access to scalar response amplitudes on each trial, which themselves displayed high trial-to-trial variability.

We next asked how a model provided with different subsets of optic glomeruli performed on the decoding task by training the model using only responses from a single functional group (identified in *Figure 3*). As expected, decoding models with access to responses from only a subset of the population performed more poorly than those with access to the full glomerulus population. Strikingly, however, subpopulations of glomeruli were unable to perform as well as the full population even for correctly classifying the stimuli to which they were most strongly tuned (*Figure 3F*). For example, group 1 contains the glomeruli that showed strong responses to small, 5° spots. Yet a model trained using the responses from that group alone was unable to encode information about this stimulus nearly as well as the full population model. To test whether similar distributed representation exists for a reduced subset of stimuli that are known to be discriminable by flies, we repeated this analysis focusing only on four stimuli that drive distinct visual behaviors: a drifting grating, a 15° spot, a looming spot, and a vertically oriented bar (*Figure 4—figure supplement 3*). As expected because the task is easier, overall decoding performance across the population was higher. As with the larger stimulus set, many stimulus classes could be decoded at above chance level by multiple glomeruli groups, and for some stimuli, like the drifting grating and vertically oriented bar, decoding ability across the population was higher than for any individual group.

In other sensory systems, positive correlations in neural responses can mitigate the effects of trial-to-trial variability in cases of heterogeneous population tuning (*Averbeck and Lee, 2006*; *Franke et al., 2016*; *Romo et al., 2003*; *Zylberberg et al., 2016*). This is because, relative to uncorrelated variability, correlated variability can be oriented in a direction in population response space where it does not interfere with stimulus decoding (see Discussion and *Pruszynski and Zylberberg, 2019*). We therefore hypothesized that the strong trial-to-trial gain correlations (*Figure 4C*) were partly responsible for the high decoding performance for some stimuli in spite of the high response variance. To test this, we trained and tested the decoding model using trial-shuffled responses, such that for each glomerulus the mean and variance of each stimulus response were the same, but the trial-to-trial correlations were removed (*Figure 4G*, left). With trial-to-trial correlations removed, the decoding model performed about 35% worse than the model trained on correlated single-trial responses (*Figure 4G*, right). The decrease in performance upon trial shuffling was present across stimuli, indicating that this is a general feature of stimulus encoding for this population, and not specific for selected visual features (*Figure 4H*). This result highlights the importance of performing simultaneous measurements to characterize population responses: using independent measurements and assuming uncorrelated response variability in this case would suggest a significantly worse single-trial decoding ability than is present in the full population. Taken together, these results show that, rather than a single visual feature being encoded by one or a few VPNs, all visual features are likely represented jointly across the population. Moreover, positive correlations in response variance enhance stimulus decoding relative to uncorrelated variability.

## Walking behavior selectively suppresses responses of small-object detecting glomeruli

Because sensory neural activity has been shown to be modulated by behavior in flies (*Chiappe et al., 2010*; *Fenk et al., 2021*; *Kim et al., 2015*; *Strother et al., 2018*; *Kim et al., 2017*) and other animals (*Maimon, 2011*; *Niell and Stryker, 2010*), we wondered whether the trial-to-trial gain changes shown above were related to the behavioral state of the animal. To test this, we measured glomerulus

population responses while the animal walked on an air-suspended ball (*Figure 5A–B*, see Materials and methods). Under this fictive walking paradigm, forward and rotational velocity components of movement were highly correlated. Because of this, we cannot disambiguate between contributions from forward and rotational velocity components in isolation. In sum, the fictive walking data show intermittent bouts of walking activity, and these movement bouts consisted of both forward and rotational velocity components. Because of this, we used total ball rotation as a measure of locomotor activity. To simplify the gain characterization, we showed a repeated probe stimulus on every trial, for 100 trials. First, we showed a 15° dark moving spot, since this stimulus drives strong responses in many glomeruli, including LC11, LC21, LC18, LC6, LC26, LC17, LC12, and LC15. We will refer to these glomeruli as 'small object detecting glomeruli', recognizing that they also respond to other stimuli (*Figure 3*). Examining the single-trial responses to the probe alongside fictive walking behavior revealed a striking relationship: probe stimuli that appeared when the fly was walking drove much weaker responses in some glomeruli than stimuli that appeared while the fly was stationary (*Figure 5C and D*). On average, responses of the LC11, LC21, L18, LC12, and LC15 glomeruli showed significant negative correlation with behavior. Conversely, responses of the LC6, LC26, and LC17 glomeruli did not show significant negative correlation with behavior. We next examined a measure of the population response gain of these five modulated glomeruli as a function of walking amplitude, across all trials and all flies (*Figure 5—figure supplement 1*). This analysis revealed that the weakest walking amplitudes were not associated with gain changes, while walking amplitudes that exceeded ~10°/s showed lower-than-average response gain.

Because glomerulus responses in these experiments are dominated by the VPN that provides most of their input (*Figure 3*), we expected that this gain modulation was due to changes in VPN responses. To test this more directly, we repeated this experiment using a specific split-Gal4 driver line for LC11 VPNs. We observed a similar negative correlation between response gain and walking amplitude using this genetically targeted approach (*Figure 5—figure supplement 2*).

We next tested whether a similar behavioral modulation exists for those glomeruli which respond more strongly to loom, namely LC6, LC26, LC16, LPLC2, LC4, LPLC1, LC9, LC17, and LC12, using a dark looming spot as a probe (*Figure 5E*). Across animals, we saw no significant modulation of loom responses by walking (*Figure 5F*). Thus, walking behavior selectively suppressed the visually evoked responses of specific optic glomeruli, with the strongest effects on a subset of small object detecting glomeruli, while having no significant effect on glomeruli that respond most strongly to loom.

The gain changes associated with walking strongly resemble the correlated gain changes we saw in earlier experiments with the broader stimulus suite (*Figure 4*). This suggests that the trial-to-trial shared gain was associated with the behavioral state of the animal. To test this idea, we examined the subset of flies from the experiments in *Figure 4* where we also collected walking behavior. We found that for each fly, the first principal component of the population response, corresponding to the large shared gain factor, was negatively correlated with walking (*Figure 5—figure supplement 3*), with an average rank correlation coefficient of $\rho=-0.23$. Thus, the shared gain modulation is associated with walking, but importantly, this relationship is incomplete. This means that one could not infer the population correlation structure seen in *Figure 4* by leveraging information about walking behavior.

## Visual inputs associated with self-generated rotation modulate glomerulus sensitivity

Self-generated motion is associated with characteristic visual cues, including wide-field, coherent visual motion on the retina. In the next series of experiments, we set out to test the hypothesis that optic glomerulus gain might be modulated by these visual signatures of self-generated motion. To test whether glomeruli respond to visual cues characteristic of walking, we first created a complex visual stimulus designed to include several features thought to be components of natural visual inputs to walking flies, including objects at different depths (vertically oriented, dark bars), as well as images dominated by low spatial frequencies (*Figure 6A*). To move this scene, we measured fly walking trajectories using a 1 m² arena with automated tracking, as described previously (*York et al., 2022*), and applied short segments of these walking trajectories to the camera location and heading in our visual environment, creating an open loop 'play-back' stimulus (*Figure 6B*). These VR stimuli drove very weak responses across all glomeruli, including the small object detecting glomeruli (*Figure 6C*), despite these glomeruli in the same flies responding very robustly to isolated vertical bars similar to

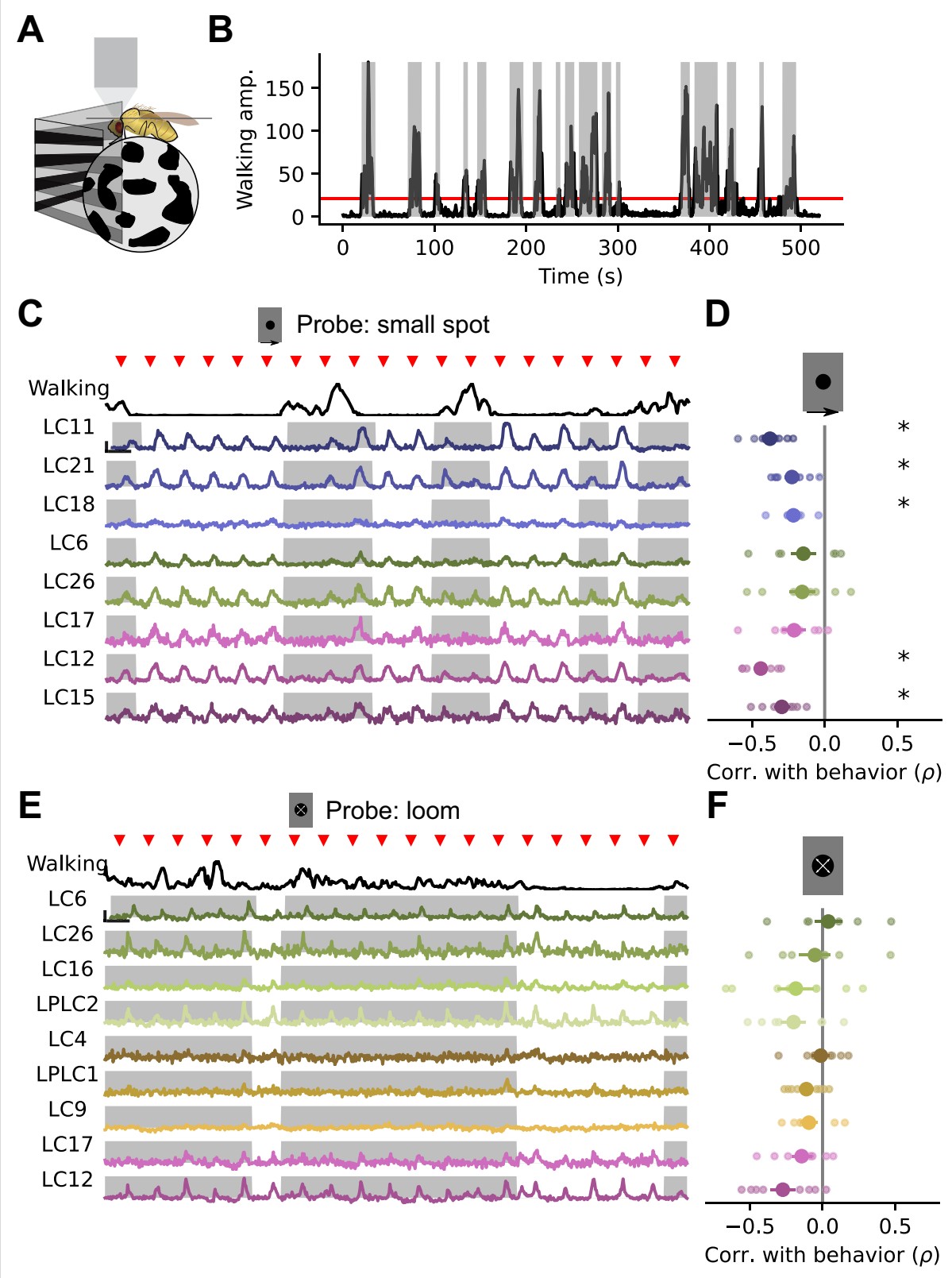

**Figure 5.** Walking behavior suppresses glomerulus responses to small object stimuli. (**A**) Schematic showing fly on air-suspended ball for tracking behavior. (**B**) We used the change in ball rotation to measure overall walking behavior. Red line indicates threshold for binary classification of behaving versus nonbehaving, and gray shading indicates trials that were classified as behaving. (**C**) We presented a 15° moving spot repeatedly to probe glomerulus gain throughout the experiment. Example responses and quantification are shown only for glomeruli which respond reliably to this probe

*Figure 5 continued on next page*

Figure 5 continued

stimulus. Red triangles show stimulus presentation times. Black traces (top) show walking amplitude, and gray shading indicates trials classified as behaving. Scale bar is 4 sec and 25% dF/F. (**D**) Correlation (Spearman's rho) between behavior and response amplitude. Each small point is a fly, large point is the across-fly mean (n=8 flies), and asterisks indicate glomeruli with a significant negative correlation between response gain and behavior (One sample t-test, p<0.05 after correction for multiple comparisons). (**E–F**) Same as C-D for a looming probe stimulus, with responses from the subset of glomeruli that respond strongly to looming stimuli. On average, there is no correlation between behavior and loom response strength (n=8 flies).

The online version of this article includes the following figure supplement(s) for figure 5:

**Figure supplement 1.** Population response gain as a function of walking amplitude.

**Figure supplement 2.** Genetically targeted VPN imaging shows behavioral modulation of response gain.

**Figure supplement 3.** The shared glomerulus gain factor is negatively correlated with behavior.

those in the scene (*Figure 6D*). The relatively weak responses of most glomeruli to these play-back stimuli suggested that some features characteristic of visual inputs during walking suppress glomerulus responses via the visual surround of each VPN. To test this idea, we focused on one prominent component of visual inputs during self-motion, namely the coherent visual rotation associated with body turns. We note that although we did not address it here, other components of self-motion, including forward translation, may also play an important role in shaping visual responses. To explore the spatial and temporal frequency tuning of this visual surround, we presented a 15° dark spot, a probe stimulus that many glomeruli respond to (*Figure 3*), while drifting a sine wave grating in the background with variable spatial period and speed (*Figure 6E*). The LC11 glomerulus, which responds strongly to small moving objects on uniform backgrounds, showed strongly suppressed probe responses to gratings with low spatial frequencies, and across speeds chosen to span the typical range of angular speeds experienced during fly locomotor turning (*Figure 6F*). This suppression by low spatial frequency gratings across a range of rotational speeds was seen for all small object detecting glomeruli (*Figure 6G*). Thus, these glomerulus responses are subject to a suppressive surround that is sensitive to low spatial frequencies and to a broad range of retinal speeds.

While the previous experiments show that the surround is responsive to low spatial frequency drifting gratings, they do not test whether the surround is selective for rotational motion. We next designed a stimulus to test this idea. A prominent feature of self-generated visual motion, especially rotational turns, is widefield motion coherence. That is, when an animal turns, all local motion signals across the visual field are aligned along an axis defined by the axis of rotation. From a visual circuit perspective, coherent rotational motion concentrates activity of elementary motion detecting neurons T4/T5 within a single layer of the lobula plate, where as incoherent local motion would spread T4/T5 activity across all layers. To test whether motion coherence impacted surround suppression of optic glomeruli, we designed a stimulus inspired by random dot kinematograms (*Britten et al., 1992*). This stimulus was composed of a field of small dots, roughly 15° in size, that moved around the fly at constant speed. Individual spots of this size drive robust responses in most small object detecting glomeruli, and should also recruit elementary motion detectors T4/T5.

This moving dot field had a tunable degree of coherence, such that at a coherence level of 0, each dot moved at the defined speed, but in a random direction. At a coherence level of 1, every dot moved in the same direction (*Figure 6H*). Intermediate coherence values correspond to the fraction of dots moving along the pre-defined 'signal' direction. Importantly, this stimulus has the same overall mean intensity, contrast and motion energy for every coherence level. As expected, at 0 coherence, small object detecting glomeruli responded strongly. However, as the motion coherence was increased, responses of many small object detecting glomeruli decreased (*Figure 6H, I*). Taken together, these results are strong evidence that the suppressive surround of these glomeruli is sensitive to widefield motion cues that are characteristic of self-motion.

## Natural images recruit surround suppression

To test whether rotational self-motion cues derived from natural scenes can drive surround suppression in small object detecting glomeruli, we used a moving 15°spot to probe response gain while presenting natural images in the background (*van Hateren and van der Schaaf, 1998*; *Figure 7A*). When presented on top of a stationary image, the probe stimulus elicited a large response in LC11. However, when the probe was presented on top of a rotating natural image, LC11 glomerulus responses

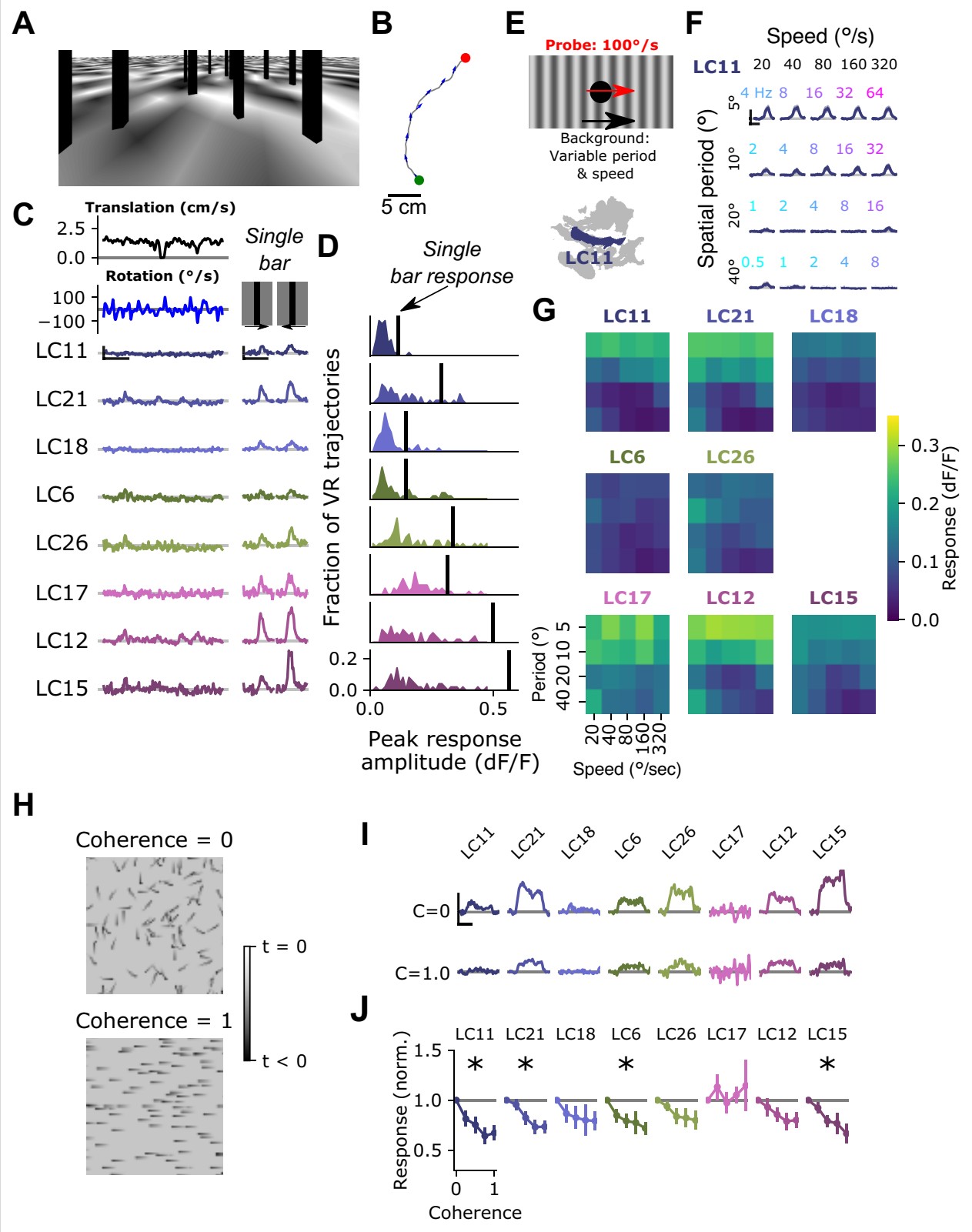

**Figure 6.** Visual input associated with rotational motion suppresses optic glomerulus responses. (**A**) Still image from a VR stimulus. (**B**) Twenty second snippet of a fly walking trajectory. Green and red points indicate start and end of the trajectory, respectively. Arrows show the fly's heading. (**C**) For an example fly, responses of small object detecting glomeruli to an example virtual reality trajectory. These glomeruli respond very weakly to the virtual reality stimulus, despite showing strong, reliable responses to solitary vertical bars (right) similar to those present in the virtual reality scene. Scale

*Figure 6 continued on next page*

*Figure 6 continued*

bar is 5 s and 25% dF/F. (**D**) Histograms showing, for each glomerulus, the distribution of peak responses to each VR trajectory. Vertical line indicates mean response to a single dark, vertical bar stimulus (five VR trajectories were presented to each fly, n=10 flies). (**E**) Schematic showing the surround suppression tuning stimulus. A small dark probe stimulus moves through the center of the screen while a grating with varying spatial frequency and speed moves in the background. (**F**) For the LC11 glomerulus, probe responses are suppressed by low spatial frequency gratings across a range of speeds consistent with locomotor turns. Small, color-coded numbers indicate the temporal frequency associated with each grating speed and spatial period. Scale bar is 2 s and 25% dF/F. (**G**) Heatmaps showing probe responses as a joint function of background spatial period and speed for each of these 8 glomeruli (n=10 flies). (**H**) Schematic of random dot coherence stimulus. At zero coherence (top image), each dot moves independently of all the other dots. At a coherence of 1.0 (bottom image), all dots move in the same direction. (**I**) For an example fly, responses of the small object detecting glomeruli are shown to coherence values of 0 (top row) and 1.0 (bottom row). Scale bar is 2 s and 25% dF/F. (**J**) Summary data showing response amplitude (normalized to the 0 coherence condition within each fly) of each glomerulus to varying degrees of motion coherence. Asterisk at the top indicates a significant difference between the response to 0 and 1 coherence (n=11 flies, paired t-test, p<0.05 after correction for multiple comparisons).

were strongly suppressed for rotational speeds spanning the range of locomotor turns (*Figure 7B*), compared to a stationary image background. In agreement with *Figure 6H–J*, this suggests that rotational motion recruits surround suppression. We next explored surround speed tuning across all eight small object detecting glomeruli (*Figure 7C*). The LC11, LC21, and LC18 glomeruli showed strong suppression at all non-zero image speeds tested (*Figure 7C*, left). The LC6 and LC26 glomeruli showed a shallower dependence of surround suppression on image speed (*Figure 7C*, center), while the LC17, LC12, and LC15 glomeruli showed intermediate speed dependence (*Figure 7C*, right). In summary, natural images suppress small object responses in these glomeruli and surround suppression generally increases with increasing background speed.

We hypothesized that the low spatial frequency content of natural images was critical for these effects, since the grating results (*Figure 6*) showed the strongest suppression for low spatial frequency gratings, and because natural images are characterized by long-range intensity correlations and low spatial frequencies (*Figure 7D*). To test the effect of spatial frequency content of images on surround suppression, we repeated this experiment with filtered versions of the natural images. For each of three natural images, we presented the original (unfiltered) image, a whitened natural image, which has a roughly flat power spectrum at low spatial frequencies, a high-pass filtered image, and a low-pass filtered image (*Figure 7D*). For LC11, and all other small object detecting glomeruli, the natural image and its low-pass filtered version strongly suppressed responses to the probe, whereas the whitened and high-pass filtered images recruited much weaker suppression (*Figure 7E and F*). We note that this spatial and temporal frequency tuning of these suppressive surrounds is broadly consistent with the tuning properties of elementary motion detecting neurons T4 and T5 (*Leong et al., 2016*; *Maisak et al., 2013*). These observations raise the possibility that local motion detectors provide critical input to the visual surrounds of small object detecting glomeruli. Moreover, the observation that surround speed tuning was similar for glomeruli that clustered together using the synthetic stimulus suite (*Figure 3*) suggests that this functional clustering may reflect properties of the surround. More broadly, the differential speed sensitivities of these surrounds may further diversify feature selectivity across these groups of glomeruli in the context of natural visual inputs.

## Behaviorally and visually driven suppression independently modulate small object detectors

The results presented thus far show that the gain of small object detecting glomeruli was tuned by both locomotor behavior and widefield visual motion. Both of these cues are associated with self-generated movements of the animal. How can the fly reliably track external objects during self-motion if small-object detecting glomeruli are suppressed by visual and behavioral cues? We hypothesized that the answer might lie in the temporal dynamics of locomotor behavior. Natural fly walking behavior is saccadic, interspersing fast turns with periods of relatively straight walking bouts (*Cruz et al., 2021*; *Geurten et al., 2014*; *Juusola et al., 2017*; *Reynolds and Frye, 2007*). We hypothesized that the saccadic structure of walking ensures that glomerulus gain is suppressed only transiently during a saccade, and once the saccade is over, visual response gain is restored to sample external objects.

To test this idea, we first examined the temporal dynamics of locomotor turns under conditions where animal movement is unconstrained. To do this, we examined walking trajectories from our open behavioral arena (see *Figure 6* and *York et al., 2022*; *Figure 8*). Walking trajectories in an open arena

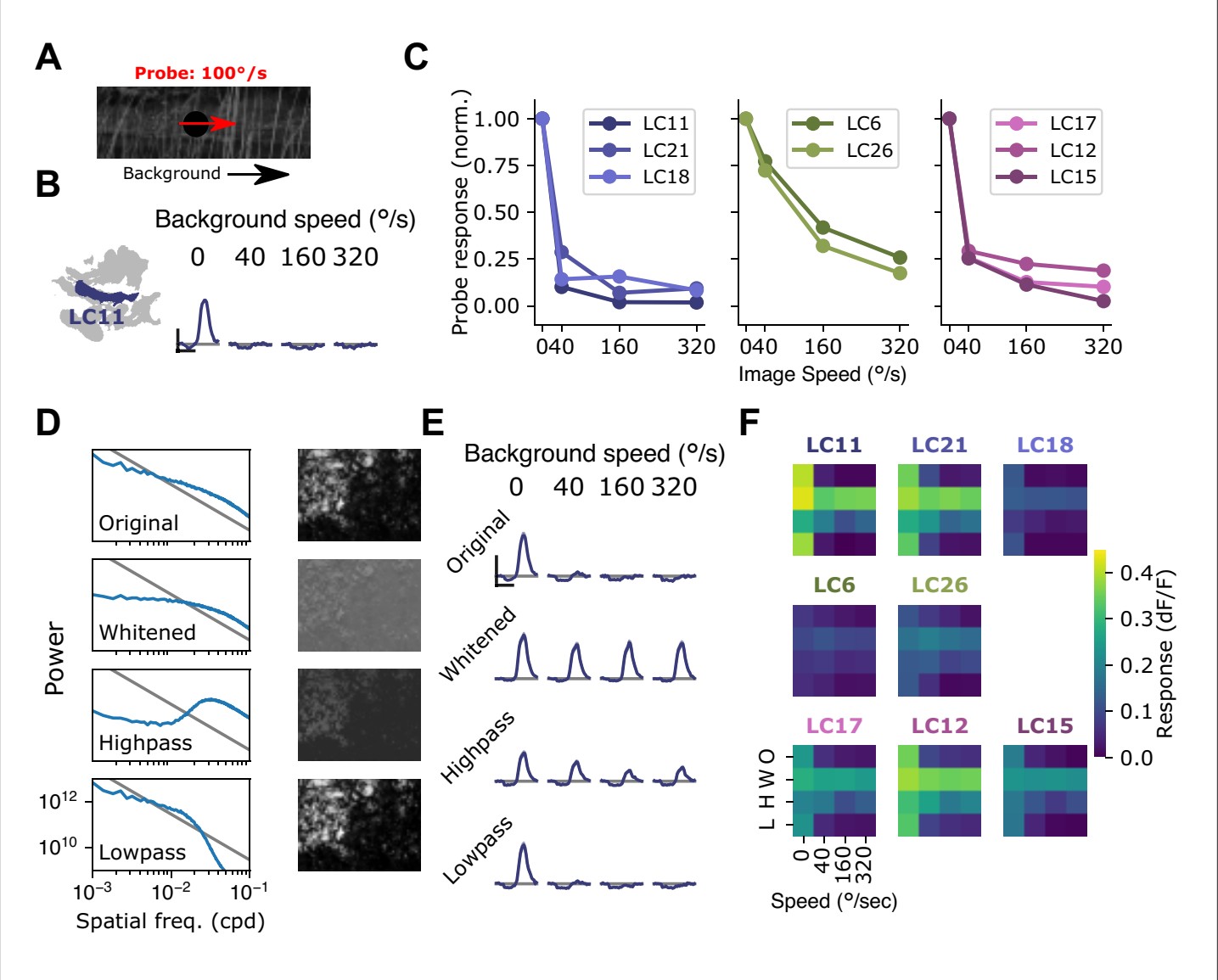

**Figure 7.** Visual suppression is tuned to natural image statistics. (**A**) Stimulus schematic: a small spot probe is swept across the visual field while a grayscale natural image moves in the background at a variable speed. (**B**) For the LC11 glomerulus, natural image movement strongly suppresses the probe response across a range of speeds (average across three images, n=8 flies). (**C**) Mean probe response as a function of background image speed, normalized by probe response with static background. Surround speed tuning curves are grouped by functional glomerulus groupings for the small object detecting glomeruli. (**D**) Average power spectra (left) and example image (right) for the original natural images (top), whitened images (second row), high-pass filtered images (third row), and low-pass filtered images (bottom row). Gray line shows p $\propto$ 1/f². (**E**) For LC11, image suppression of probe responses is attenuated by whitening the image or high-pass filtering it, but not by low-pass filtering the image. (**F**) Dependence of probe suppression on image speed and filtering for each of the eight small object detecting glomeruli (n=9 flies). Scale bar is 2 s and 25% dF/F.

showed a wide range of angular speeds and forward velocities (*Figure 8A and B*). Examining the angular velocity of a single walking trajectory revealed saccadic temporal dynamics (*Figure 8C*). Across all flies, the time between saccades (the 'inter-turn interval') showed a wide range, but there were few inter-turn intervals less than ~0.5 s, a peak near 1 s, and a long tail (*Figure 8D*). We chose a threshold angular speed to classify saccadic turns, here 160°/s, but inter-turn interval distributions were similar across a range of threshold values that include the vast majority of turns. For all saccade thresholds, there was a low probability of a saccade within ~0.5 s of the previous saccade. We next split up snippets of walking velocity trajectories based on whether they occurred within a 400 ms window around a saccade or during an intersaccade interval. As expected, angular speeds experienced during a

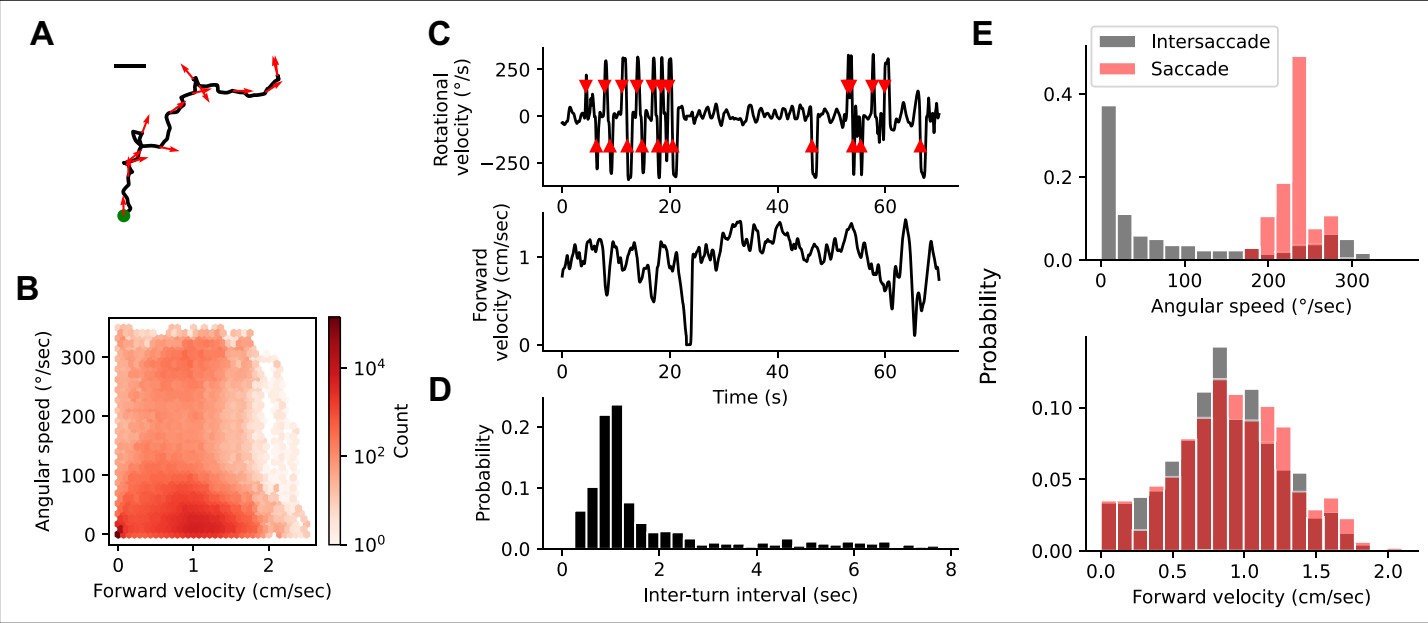

**Figure 8.** Natural fly walking is punctuated by saccadic turns. (**A**) Example unconstrained fly walking trajectory measured in an open behavioral arena. Arrows show the fly's heading. Green point indicates start of walking trajectory. Scale bar=1 cm. (**B**) Heatmap of instantaneous forward velocity and rotational speed across all time points from 81 fly trajectories. Note logarithmic color scale. (**C**) Example trace of instantaneous angular velocity (top) and instantaneous forward velocity (bottom). Red arrowheads on top indicate saccades. (**D**) Histogram of inter-turn intervals shows a peak around 1 s and a long tail. (**E**) Histogram of angular speed (top) and forward velocity (bottom) within a 400 ms window around a saccade (red) or during the intersaccade interval (gray).

saccade were large, and during intersaccade intervals, most angular speeds were very low (**Figure 8E**, top). Interestingly, distributions of forward velocities were not different during a saccade compared to the intersaccade interval (**Figure 8E**, bottom), meaning that saccades tend to occur while the fly is moving forward as well. Taken together, this means that a typical locomotor saccade is followed by at least a 500 ms, and often a ~1-s period of relative heading stability.

We next asked whether saccadic visual inputs recruit surround suppression, and whether the timescale of this suppression could support such a visual sampling strategy. We designed a stimulus meant to mimic the retinal input during a locomotor saccade. As before, we presented a probe stimulus on every trial to measure the response gain of small object detecting glomeruli. In the background was a grayscale natural image (**Figure 9A**), which underwent a lateral rotation of 70° in 200 ms at a variable time relative to the glomerular response to the probe (**Figure 9B**). As a result, the saccade signal could precede, co-occur with, or lag the glomerulus response to the probe. When the saccade occurred within ~500 ms of the probe response, the probe response was attenuated, suggesting that this saccade stimulus recruits the motion-sensitive suppressive surround. Across many small object detecting glomeruli, including LC11, LC21, LC17, LC12, and LC15, we saw strong gain suppression when the saccade occurred around the time of the probe response (**Figure 9C and D**). Interestingly, the other glomeruli, LC18, LC6, and LC26 showed much weaker and more variable saccade suppression, maintaining their response gain regardless of saccade timing. Note that because we presented only one saccade on each probe trial, and because we are quantifying gain using the response amplitude, the timing dependence measured here is independent of calcium indicator dynamics and therefore reflects dynamics associated with the glomerulus response. The timescale of this surround suppression, combined with the temporal dynamics of fly turning (**Figure 8**) suggests that visually driven saccade suppression transiently reduces glomerulus response gain around the time of a locomotor saccade, but gain recovers while the fly's heading is stable and before the next saccade occurs. We infer that this dynamic gain adjustment allows the fly to sample the scene during the inter-saccadic periods of heading stability.

Because the brief saccade stimulus did not completely suppress probe responses, we could examine the relationship between visual-related and motor-related gain control mechanisms. One

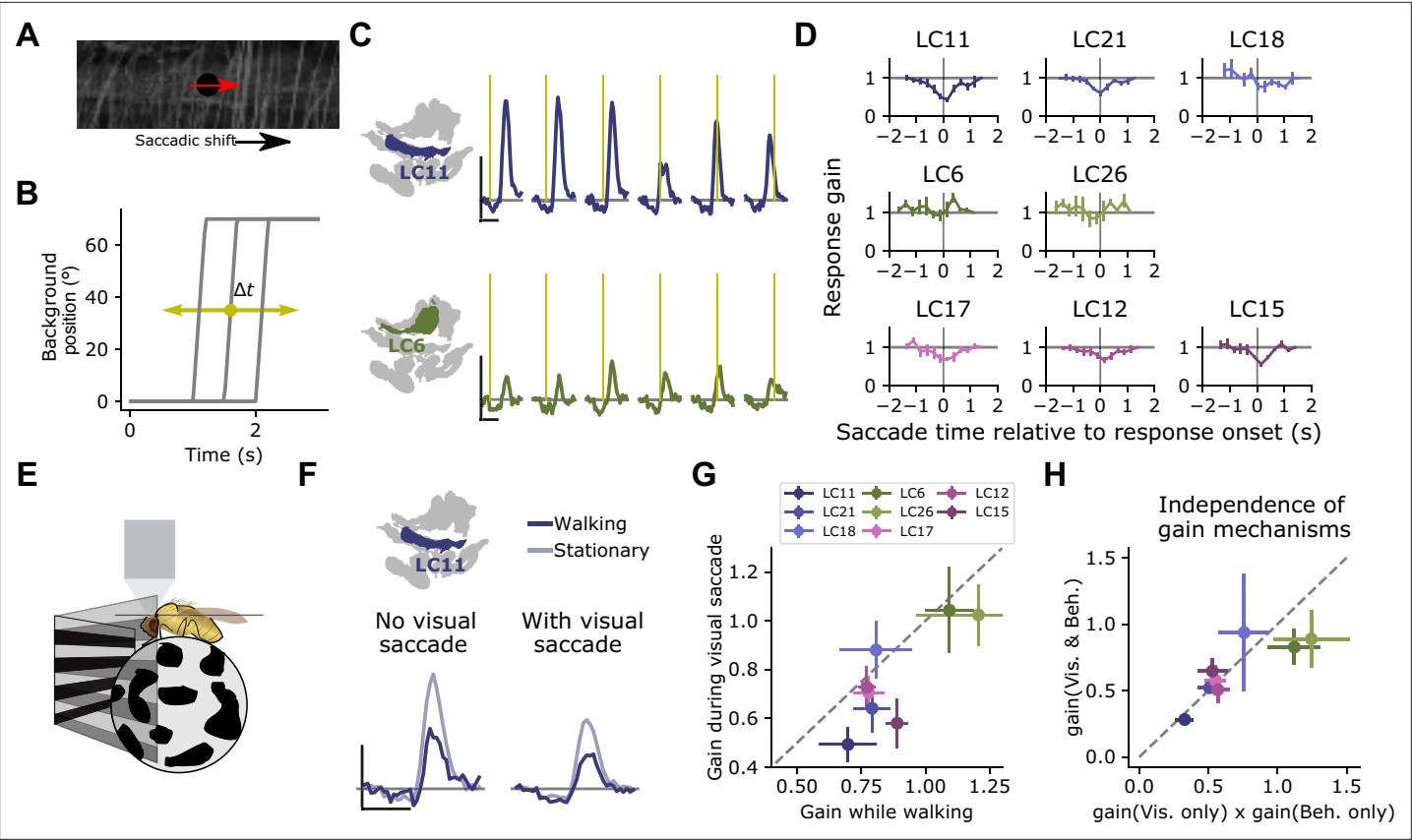

**Figure 9.** Visually driven saccade suppression works in concert with behavioral suppression. (**A**) Stimulus schematic: A small moving probe is used to measure glomerulus response gain while a full-field natural image in the background undergoes a fast, ballistic lateral displacement meant to mimic fly walking saccades. (**B**) The time of the saccade is varied throughout the trial, such that it occurs at different phases of the probe response. Each saccade lasted 200 ms and translated the image by 70°. (**C**) Trial-average probe responses in an example LC11 glomerulus (top) and an example LC6 glomerulus (bottom). Saccade times are indicated by the yellow vertical line in each panel. As the saccade approaches the probe in time, the probe response is suppressed in LC11 but not LC6. (**D**) Summary data showing, for each small object detecting glomerulus, the response gain as a function of saccade time relative to the response onset. t=0 corresponds to coincident probe response and saccade onset. Suppression is strongest when the saccade occurs near probe response onset (n=5 flies). (**E**) We monitored fly walking behavior while presenting the saccadic visual stimuli above. (**F**) For an example LC11 glomerulus, both the saccadic stimulus and walking behavior reduced probe responses, and these gain control mechanisms could be recruited separately. (**G**) Population data showing average response gain for visual saccades versus response gain during walking behavior, for each small object detecting glomerulus. Most glomeruli lie below unity (dashed line), indicating that visual saccade suppression is typically stronger than behavior-linked suppression. These two sources of suppression are also correlated across glomeruli (r=0.80, Pearson correlation coefficient). (**H**) For each small object detecting glomerulus, we compared the response gain when both visual-related and motor-related gain modulation were recruited (vertical axis) to the product of both response gains in isolation (horizontal axis). Dashed line is unity, and points falling along that line indicate a linear interaction between those forms of gain control.

way to interpret these data is that motor signals suppress small object detecting glomeruli, and that widefield, coherent visual motion induces a turning response, recruiting the same motor-command derived suppression. Is the apparent visual suppression a result of the motor feedback, or are the visual-based and motor-based suppression mechanisms independent? To test this idea, we monitored walking behavior while presenting saccadic visual stimuli (**Figure 9E**). We first examined the LC11 glomerulus response to the probe under both behavioral conditions (walking versus stationary), and under both visual conditions (saccade coincident with the probe response, vs. no saccade coincident with the probe response). Strikingly, when the fly was stationary, visual saccades still reduced the gain of the response (**Figure 9F**). Interestingly, the glomeruli that were subject to stronger gain reductions by the visual saccade also showed stronger gain reductions by walking (**Figure 9G**, r=0.80). Finally, to test whether these two gain modulation mechanisms were independent, we compared the measured probe responses when the fly was receiving both saccadic visual input and was walking to the product of each gain change measured independently (i.e., when the fly was receiving either

saccadic visual input or was walking, but not both) (*Figure 9H*). Across the small object detecting glomeruli, the prediction that these two gain mechanisms were independent accurately captured the jointly measured gain modulation. Taken together, these data indicate that visual suppression is not the indirect effect of an induced turning response and that saccadic visual modulation and behavior-related modulation are both balanced in magnitude and independent.

## Discussion

In this study, we show that local feature detection is challenged by rotational self-motion signals in rich visual environments (*Figure 1*). To determine how feature detecting neurons might maintain selectivity under natural viewing conditions, we first developed a new connectome-based method to segment functional imaging signals that allowed us to measure neural responses across a heterogeneous population of VPNs (*Figures 2–3*). Using this method, we found that information about different visual features is distributed across multiple VPN types, meaning that stimulus identity cannot be decoded from a single glomerulus alone (*Figure 4*). Further, we found that strong trial-to-trial response correlations improve stimulus encoding fidelity (*Figure 4*). Strikingly, the locomotor behavior of the fly selectively modulated responses of small object detecting, but not loom detecting, glomeruli (*Figure 5*). We then showed that visual motion signals characteristic of walking also modulated the responses of glomeruli tuned to small objects (*Figures 6–7*). Finally, we demonstrated that visual suppression occurs during naturalistic body saccades made by walking flies, and that behavioral and visual gain modulation are both balanced in magnitude and independent, such that these two cues combine linearly (*Figures 8–9*). Taken together, these two forms of gain control reduce the sensitivity of small object detectors to inputs that can diminish the discriminability of local features, thereby allowing for reliable feature detection during saccadic vision.

### Population coding of local visual features

Our characterization of the optic glomeruli using solitary visual features (*Figure 3*) largely agrees with what has been described previously, in that many glomeruli respond strongly to small moving objects, others respond to visual loom, and responses to stationary flicker or widefield motion are weak or nonexistent (*Hindmarsh Sten et al., 2021*; *Keleş and Frye, 2017*; *Keleş et al., 2020*; *Klapoetke et al., 2022*; *Städele et al., 2020*). These data have been used as evidence that particular VPNs are linked to specific visual features and corresponding visually guided behaviors (*Hindmarsh Sten et al., 2021*; *Ribeiro et al., 2018*). At the same time, the responses of individual VPN classes overlap, in the sense that an individual visual stimulus will evoke responses from many VPN classes, suggesting a dense population code. We note, however, that how downstream circuits make use of the information available across VPNs to guide behavior is not well understood, and further work incorporating connectomics, targeted perturbations of VPN channels, and behavioral analyses might shed light on this question. For example, a recent study used genetic silencing, coupled with a goal-oriented neural network model, to show that VPNs jointly encode behaviorally relevant visual features during *Drosophila* courtship (*Cowley et al., 2022*).

To what extent is it possible to decode stimulus identity based on the activity of a single VPN class? Our results show that population measurements are important to describe feature encoding by VPNs for two reasons. First, evaluation of stimulus decoding revealed that most visual features are encoded jointly across the population, not by single VPN types. This is because information about stimulus identity is contained not only in the responses of glomeruli that are strongly tuned to a particular feature but also in the weaker responses of glomeruli that have different tuning properties. Second, while responses in each VPN type showed high trial-to-trial variability, simultaneous measurements revealed that this variability was strongly correlated across the population, and that this improved coding fidelity across the population relative to uncorrelated variability. This is consistent with past experimental and theoretical work showing that positive correlations can help offset the deleterious effect of response variability by shaping the noise in directions in population response space that do not interfere with stimulus decoding (*Franke et al., 2016*; *Zylberberg et al., 2016*; *Pruszynski and Zylberberg, 2019*; *Averbeck and Lee, 2006*; *Moreno-Bote et al., 2014*). A similar structure of neural variability relative to population tuning permits accurate stimulus decoding in the face of large movement-related signals in mouse cortex (*Rumyantsev et al., 2020*; *Stringer et al., 2021*). This

effect relies on heterogeneous tuning across the population of neurons, and downstream decoders can extract information about the stimulus by comparing activation across differently tuned neurons. Indeed, in the case of a population of identically tuned neurons, positive noise correlations degrade rather than improve coding fidelity (*Zohary et al., 1994*; *Averbeck et al., 2006*).

The trial-to-trial variability we observed among VPNs was dominated by a single, shared population response gain that was associated with walking behavior, but only weakly (*Figure 5—figure supplement 3*). Thus, this shared gain is likely modulated by other factors, for example, shared upstream noise (*Ala-Laurila et al., 2011*; *Zylberberg et al., 2016*) or other behavioral or physiological states that we did not measure. How downstream circuits combine signals across glomeruli may provide insight into how the brain decodes VPN population responses to encode local features, and available connectomic data sets can accelerate progress on this question (*Klapoetke et al., 2022*; *Scheffer et al., 2020*).

## Natural locomotor behavior modulates the sensitivity of small object detectors

Behavior-associated gain changes are widespread in visual systems across phyla (*Maimon, 2011*; *Maimon et al., 2010*; *McAdams and Maunsell, 1999*; *McBride et al., 2019*; *Niell and Stryker, 2010*). Recent work demonstrates that locomotor signals are prevalent throughout the *Drosophila* brain, including in the visual system (*Aimon et al., 2019*; *Brezovec et al., 2022*; *Schaffer et al., 2021*), but has been examined most extensively in circuits involved in elementary motion detection and widefield motion encoding. Behavioral activity has been shown to modulate response gain in widefield motion detecting lobula plate tangential cells (LPTCs) and some of their upstream circuitry (*Chiappe et al., 2010*; *Kohn et al., 2021*; *Maimon et al., 2010*; *Strother et al., 2018*; *Suver et al., 2012*), and LPTC membrane potential tightly tracks walking behavior, even in the absence of visual stimulation (*Fujiwara et al., 2017*; *Fujiwara et al., 2022*). During flight, efference-copy based modulation of LPTC membrane potential has been proposed to cancel expected visual motion due to self-generated turns (*Fenk et al., 2021*; *Kim et al., 2015*; *Kim et al., 2017*). In each of these cases, behavioral signals adjust response gain according to expected visual inputs, for example faster rotational speeds during flight. *Kim et al., 2015* showed that flying saccades were associated with hyperpolarization of optic glomeruli interneurons, which the authors speculated could be useful for canceling spurious small object detector responses during self-motion, much like what we see in VPNs during walking behavior. The behavioral gain modulation we describe here selectively adjusts visual sensitivities to reflect the fact that specific visual inputs are particularly corrupted by self-motion. Small object detection is an especially challenging task during self-motion (*Figure 1*), and consequently, gain modulation most strongly affects glomeruli involved in this task. Glomeruli that are tuned more strongly to looming visual objects were not modulated by walking behavior, suggesting that these larger visual features can be reliably extracted under walking conditions. During a high-velocity locomotor saccade, sensitivity to small objects is transiently decreased, and in the subsequent inter-saccade interval, small object detector gain is restored, allowing for selective encoding of visual features at different points in the locomotor cycle.

Interestingly, our estimates of visual gain suppression (*Figure 7*) combined with the statistics of free walking (*Figure 8*), suggest that some small object detecting glomeruli may experience visual suppression due to the smaller rotational motion present during intersaccadic periods, as well. For example, LC11, LC21, and LC18 showed strong visual suppression even for image rotations of only 40°/s, which can be achieved in the time between saccades. This suggests that these cells operate best as small feature detectors during periods of high heading stability, which may be achieved during some periods of straight, forward walking or while the fly is stationary. We have chosen to focus on visual inputs during rotation because these movements cause rapid, uniform shifts in visual inputs, but we note that the more complex widefield motion inputs that are associated with forward translation, which produces nonuniform flow fields across the retina, likely also impact local visual feature encoding.

Importantly, our experiments measuring fictive walking on an air-suspended ball (*Figure 5*) are not able to decouple forward from rotational components of velocity, because these velocity components are highly correlated. During free walking, saccades nearly always co-occur with forward movement, making disambiguation of these walking components difficult even under natural conditions. However,

free-walking flies do perform forward walking bouts without much rotational velocity component (*Figure 8*). Further work to recapitulate natural walking statistics under conditions of head fixation would help elucidate the relative contributions of specific locomotor components to motor-related visual gain control.

Is the behavioral modulation of small object detecting glomeruli related to the well studied modulation of widefield motion detecting circuits? A parsimonious explanation of both of these observations is that neurons in the elementary and widefield motion pathways feed into the suppressive surround of small object detecting glomeruli, as is the case for figure detecting neurons in blowfly (*Egelhaaf, 1985*; *Warzecha et al., 1993*). This would endow optic glomerulus surrounds with both the widefield, coherent motion sensitivity as well as the behavioral modulation that we see. In support of this proposed mechanism, the glomeruli that show strong visual suppression are also subject to strong behavioral suppression (*Figure 9*). This hypothesis further predicts that glomeruli which derive their excitatory center inputs from elementary motion detectors (e.g., the loom-selective LPLC2; *Klapoetke et al., 2017*) might be positively gain modulated under other behavioral conditions, such as flight. Taken together, these results demonstrate that understanding local feature detection during natural vision requires accounting for the structure of locomotion. More broadly, we have shown that walking behavior modulates a subset of glomeruli, raising the possibility that different behavioral states might selectively alter other glomeruli subsets, reshaping population coding of visual features to subserve different goals.

## Motor signals and visual cues provide independent inputs to feature detectors

In addition to the motor-related gain modulation, small object detecting glomeruli are modulated by a visual surround that is tuned to widefield, coherent visual motion that would normally be associated with locomotion. This is similar to motion-tuned surrounds in object motion-sensitive cells in the vertebrate retina (*Baccus et al., 2008*; *Olveczky et al., 2003*), and in figure detecting neurons of the blowfly, which are suppressed by optic flow produced by self-motion (*Egelhaaf, 1985*; *Kimmerle and Egelhaaf, 2000*). Why would the fly visual system rely on these two seemingly redundant cues to estimate self-motion? One possibility is that either cue alone could be unreliable or ambiguous under some conditions. For example, a striking characteristic of natural scenes is their immense variability from scene to scene. As a result, detecting small moving objects could occur against a background of a dense, contrast-rich visual environment like a forest or a uniform, low-contrast background like a cloudy sky. These two scenes would be expected to be associated with very different wide-field motion signals, even given the same self-motion. Because of this, relying on visual cues alone for evidence of self-motion will be unreliable under the diversity of natural scenes. Thus, motor signals and visual cues characteristic of self-motion work together to provide a robust estimate of self-motion to feature detectors.

Our observation that small object detecting glomeruli are modulated by a visual surround tuned to widefield motion agrees with previous observations that flies use global motion as well as local figure information to support object tracking behavior during flight (*Aptekar et al., 2012*; *Aptekar et al., 2015*), where rotational velocities are much greater in magnitude than those associated with locomotor turns (*Fry et al., 2003*). How strategies for reliable object tracking during walking relate to flying conditions is not clear, and more work is needed to understand how small object detectors can support object tracking under these drastically different visual conditions.

## Saccade suppression as a general visual strategy

Visual motion is a prominent feature of realistic retinal inputs for both flies and vertebrates. Primates make frequent eye movements at different spatial scales during free viewing which can rapidly translate the image on the retina (*Rucci and Victor, 2015*; *Van Der Linde et al., 2009*; *Zuber et al., 1965*). Eye movements in primates are dominated by saccades, large movements that can shift the image on the retina by up to tens of degrees of visual angle. Walking flies perform locomotor saccades, which similarly rapidly shift the image impinging on the retina in a short time period (*Figure 8*; *Cruz et al., 2021*; *Geurten et al., 2014*). We found that the responses of some small object detecting glomeruli were suppressed around the time of a simulated visual saccade, while others (LC18, LC6, and LC26) showed no visual saccade suppression. Similarly, saccades in primates induce variable changes in

response gain across different brain regions, a physiological effect thought to underlie the perceptual phenomenon of saccadic suppression (*Binda and Morrone, 2018*; *Bremmer et al., 2009*; *Wurtz, 2018*; *Thiele et al., 2002*). Our data show that in flies, a similar form of saccade-related suppression can be recruited selectively to circuit elements whose feature selectivity is most sensitive to the corrupting effect of self-motion on the visual input. More broadly, this work suggests that a saccade-and-sample visual strategy is shared between flies and primates.

# Materials and methods

## Data and code availability

Data collected for this study can be found on Dryad at https://doi.org/10.5061/dryad.h44j0zpp8. All software and analysis code used for this study can be found on GitHub. Of particular note, the analysis code used to analyze these data and generate the figures presented here, can be found on GitHub at https://github.com/mhturner/glom_pop; *Turner, 2022*.

## Fly lines and genetic constructs

We generated the 20xUAS-syt1GCaMP6f construct (Addgene plasmid #190896) by cloning the cDNA sequence of *Drosophila* synaptotagmin 1, a 3× GS linker, and the GCaMP6f sequence into the pJFRC7-20XUAS vector (*Pfeiffer et al., 2010*) (Genscript Biotech). The GS linker connects the C-terminus of syt1 to the N-terminus of GCaMP6f (after *Cohn et al., 2015*). Transgenic flies were generated by PhiC31-mediated integration of the construct to produce two landing site insertions (BestGene): P{20xUAS-syt1GCaMP6f}attP40, and PBac{20xUAS-syt1GCaMP6f}VK00005. Both insertions express well. P{20xUAS-syt1GCaMP6f}attP40 was used in the paper.

The genotype of flies used for pan-glomerulus imaging was the following:

$$\frac{w^+}{w^-}; \frac{UAS-syt1GCaMP6f}{+}; \frac{UAS-myr::tdTomato}{TI\{2A-GAL4\}ChAT\{2A-GAL4\}}$$

For Split-Gal4 imaging (*Figure 3*), we used the following genotype:

$$\frac{w^+}{w^-}; \frac{UAS-syt1GCaMP6f}{LCxx-p65.AD}, \frac{UAS-myr::tdTomato}{LCxx-GAL4.DBD}$$

where LCxx corresponds to a pair of LC subtype-specific hemidrivers from *Wu et al., 2016*.

## Animal preparation and imaging

Female flies, 2–7 days post eclosion, were selected for imaging. Flies were cold anesthetized and mounted in a custom-cut hole in an aluminum shim at the bottom of an imaging chamber before being immobilized with UV curing glue. The front left leg was removed to prevent occluding the left eye, and the proboscis was immobilized using a small drop of UV curing glue. The cuticle covering the left half of the posterior head capsule was removed using a fine dissection needle, and fat bodies and trachea covering the brain were removed. The prep was continuously perfused with room temperature, carbogen-bubbled fly saline throughout the experiment. We imaged the left optic glomeruli in each fly.

For in vivo imaging, we used a two-photon resonant scanning microscope (Bruker) with a 20× 1.0 NA objective (Leica) and a fast piezo-driven Z drive to control the focal plane during volumetric imaging. Two-photon laser wavelength was 920 nm and post-objective power was ~15 mW. We collected red and green channel fluorescence to image myr::tdTomato and syt1GCaMP6f, respectively. For functional scans, to record GCaMP responses, we collected volumes with voxel resolution $1 \times 1 \times 4$ μm$^3$ (x, y, z) at a sampling frequency of 7.22 Hz. For high-resolution anatomical scans, voxels were $0.5 \times 0.5 \times 1$ μm$^3$. The imaging volume for glomerulus imaging was $177 \times 101 \times 45$ μm$^3$. Each fly was typically imaged for approximately 30–45 min. For Split-Gal4 imaging, we used the same imaging parameters that we did for the pan-glomerulus imaging experiments. Only animals with visible GCaMP6f responses in the lobula or in the optic glomeruli were included.

## Visual stimulation

We back-projected visual stimuli from two LightCrafter 4500 projectors onto a fabric screen covering the front visual field of the animal. The screens subtended approximately 60° in elevation and 140°

in azimuth. We used the blue LED of the projectors and a 482/18 nm bandpass spectral filter to limit bleedthrough into our green PMT channel. Visual stimuli were generated using a python and OpenGL-based, open-source software package we have developed in the lab, called flystim https://github.com/ClandininLab/flystim; *Steven et al., 2022*. Flystim renders three-dimensional objects in real time and computes the required perspective correction based on the geometry of the screen and animal position in the experimental setup to generate perspective-appropriate virtual reality stimuli. Rotating stimuli (e.g., gratings and images) were rendered as textures on the inside of virtual cylinders. Small spot stimuli were rendered as patches moving on cylindrical or spherical trajectories. Another custom, open-source software package, visprotocol https://github.com/ClandininLab/visprotocol, *Turner and Choi, 2022* was used to control visual stimulation protocols and handle experimental metadata.

Stimulus code for every stimulus used here can be found in the GitHub repositories for flystim and visprotocol. Below we describe some of the key visual stimulus parameters. For the synthetic visual stimulus suite, we presented 32 distinct stimulus parameterizations. All stimuli were presented from a mean gray background that remained on, between trials, throughout the entire experiment. Each stimulus presentation period was 3 s long, and was preceded and followed by 1.5 s of pre- and tail time with a mean gray background. Note that we also presented uniform flashes of ±100% contrast, but these stimuli did not drive responses in any glomerulus so we have excluded these stimuli from this paper. Visual stimuli were randomly interleaved within each imaging series.

For natural image experiments (*Figures 1, 7 and 8*), we used grayscale natural images from the van Hateren database (*van Hateren and van der Schaaf, 1998*). When presenting filtered versions of natural images, we rescaled the filtered images such that they had the same mean and standard deviation pixel values as the original images. We scaled the whitened images to have the same peak pixel intensity as the original image.

For the saccade stimulus (*Figure 9*), we used a van Hateren natural image as the background while a small, dark probe stimulus (15° in diameter) moved across the screen at 100°/s. The background image was translated by 70° over 200 ms to mimic fly walking saccades (*Cruz et al., 2021*).

Virtual reality stimuli (*Figure 6*) consisted of a 3D environment with a Gaussian-smoothed random noise texture on the 'floor' and a collection of randomly located vertical, dark, cylinders. To simulate the visual input that would be generated from *Drosophila* walking through such an environment, we moved the camera through the scene according to measured fly walking trajectories. Trajectories of female flies walking in the dark were measured in a 1 m² arena with automatic locomotion tracking, as described previously (*York et al., 2022*). About 20 s snippets from measured trajectories were selected to include periods of locomotor movement, and to exclude long stationary periods. Each fly was presented with five walking trajectories, each with its own randomly-generated pattern of cylinder locations, and five trials of each trajectory were shown.

## Behavior tracking

For experiments with behavior tracking, we raised a patterned, air-suspended ball underneath the fly to monitor its fictive walking behavior, as in *Brezovec et al., 2022*. We monitored the fly and ball movement using IR illumination and a camera triggered by our imaging acquisition software at 50 Hz frame rate.

## Alignment between in vivo functional imaging data and glomerulus map

To assign voxels in a single fly's functional in vivo image to an optic glomerulus of interest, we generated a chain of image registrations using ANTsPy (*Avants et al., 2014*; *Tustison et al., 2021*). First, each volumetric image series, including both functional and anatomical scans, was motion corrected using the myr::tdTomato signal. We then created a 'mean brain' using high-resolution anatomical scans from 11 different animals, which we aligned to one another using the myr::tdTomato channel, and averaged iteratively until a clean, crisp mean brain of the PVLP/PLP was produced. The syt1G-CaMP6f channel of the mean brain was then used to register the mean brain to a hand-cropped subregion of the JRC2018 template brain (*Bogovic et al., 2020*). To generate glomerulus masks, we first extracted the presynaptic T-bar locations in the PVLP/PLP for all LC and LPLC neurons using the *Drosophila* hemibrain connectome (*Scheffer et al., 2020*) and custom-written R code relying on the natverse suite of registration tools (*Bates et al., 2020*). We used a published transformation between

JRC2018 space and the *Drosophila* hemibrain connectome space (*Scheffer et al., 2020*), as a start to map hemibrain synapse locations to JRC2018 space, but we also computed a small additional transformation between VPN T-Bar density and JRC2018 to improve alignment at the glomerulus level. This yielded masks for each glomerulus in our in vivo mean brain space. Finally, each fly's functional image was registered to that fly's own high-resolution anatomical scan, and this anatomical scan was aligned to the mean brain. We could then bring each glomerulus mask into the functional image space of each individual fly. These masks were used to collect voxels corresponding to each distinct glomerulus, and the included voxel signals were averaged over space to yield the glomerulus response. For Split-Gal4 imaging data, we hand-drew ROIs in the glomerulus.

## Analysis of visually evoked calcium signals

Glomerulus responses from the imaging series were aligned to visual stimulus onset times using a photodiode tracking the projector timing. We used a window of time before stimulus onset (typically 1–2 s) to measure a baseline fluorescence for each trial. Using this baseline, we converted trial responses to reported dF/F values. For the functional clustering presented in *Figure 3*, we used a complete linkage criterion. Statistical significance was determined using step-down Bonferroni corrected p values from t test, and a significance criterion of 0.05.

## Small object discriminability analysis

For the small object discrimination task in *Figure 1*, we moved a 15° dark patch across a grayscale natural image and through a 'receptive field' similar in size to small object detecting VPNs. For each time point, we defined the local luminance as the average pixel intensity within the receptive field and the local spatial contrast as the variance of pixel intensities normalized by the mean pixel intensity within the receptive field. We quantified discriminability between the 'spot present' and 'spot absent' conditions using d', defined below:

$$d' = \frac{\text{mean}_{\text{present}} - \text{mean}_{\text{absent}}}{\sqrt{(\text{var}_{\text{present}} + \text{var}_{\text{absent}})/2}}$$

where mean and var represent the mean and variance of luminance or contrast within the time window when the patch passed through the receptive field. For luminance-based discrimination, we inverted the sign of d' because the presence of the patch was indicated by a decrease in local luminance.

## Single-trial stimulus decoding model

For the single-trial decoding model presented in *Figure 4*, we used a multinomial logistic regression model to predict stimulus identity using a vector of glomerulus response amplitudes for each trial. For the decoding model, responses for each glomerulus were z-scored to standardize the mean and variance across glomeruli. To train the model, we used 90% of trials, and the remaining 10% of trials were used to test performance. We iterated training/testing 100 times and we present averages across all iterations. For the trial shuffling analysis in *Figure 4*, we shuffled response amplitudes across trials of the same stimulus identity independently for each glomerulus, such that the stimulus-dependent means and variances of responses were the same, but the covariance structure was removed.

## Analysis of behavior data

To measure fictive walking behavior from video recordings of flies on an air-suspended ball, we used FicTrac (*Moore et al., 2014*) to process videos post hoc. To measure walking amplitude, at each point in time, we calculated the magnitude of the total rotation vector, using the ball rotation over all three axes of rotation, that is, walking amplitude$=\sqrt{\text{rot}_x^2 + \text{rot}_y^2 + \text{rot}_z^2}$. To classify trials as walking versus not walking, a threshold was automatically determined for each walking amplitude trajectory, using the Li minimum cross entropy method (*Li and Lee, 1993*). A trial was classified as walking if the walking amplitude exceeded this threshold for at least 25% of the time points in that trial.

## Acknowledgements

The authors thank Estela Stephenson for excellent technical support. Steven Herbst designed the original version of flystim, of which an updated version was used for visual stimulation for this work. The authors thank Fred Rieke, Karin Nordström, the reviewers, and members of the Clandinin lab for helpful feedback on earlier versions of this manuscript. This project was supported by NIH grants F32-MH118707 (MHT), K99-EY032549 (MHT), R01 EY022638 (TRC), R01NS110060 (TRC), the NSF GRFP (AK), and an NDSEG fellowship (MMP).

## Additional information

### Funding

| Funder | Grant reference number | Author |
|---|---|---|
| National Institutes of Health | F32-MH118707 | Maxwell H Turner |
| National Institutes of Health | K99-EY032549 | Maxwell H Turner |
| National Institutes of Health | R01-EY022638 | Thomas R Clandinin |
| National Institutes of Health | R01NS110060 | Thomas R Clandinin |
| National Science Foundation | GRFP | Avery Krieger |
| National Defense Science and Engineering Graduate | Fellowship | Michelle M Pang |

The funders had no role in study design, data collection and interpretation, or the decision to submit the work for publication.

### Author contributions

Maxwell H Turner, Conceptualization, Data curation, Software, Funding acquisition, Investigation, Visualization, Methodology, Writing - original draft, Writing - review and editing; Avery Krieger, Investigation, Writing - review and editing; Michelle M Pang, Resources, Investigation, Writing - review and editing; Thomas R Clandinin, Conceptualization, Supervision, Funding acquisition, Writing - original draft, Writing - review and editing

### Author ORCIDs

Maxwell H Turner http://orcid.org/0000-0002-4164-9995
Thomas R Clandinin http://orcid.org/0000-0001-6277-6849

### Decision letter and Author response

Decision letter https://doi.org/10.7554/eLife.82587.sa1
Author response https://doi.org/10.7554/eLife.82587.sa2

## Additional files

### Supplementary files
• MDAR checklist

### Data availability

All software and code is available on GitHub. Main analysis, modeling and figure generation code can be found here: https://github.com/mhturner/glom_pop, (copy archived at swh:1:rev:4a8de1aba83b-f1a7f2baadd86e23234d5cddd9fa); Visual stimulus code can be found here: https://github.com/ClandininLab/visanalysis, (copy archived at swh:1:rev:9e50cf2f38ea0e78dcab6818ff7ad0d1b7a1585a) and here: https://github.com/ClandininLab/flystim, (copy archived at swh:1:rev:bcc8f3e106544444

e3442396b14b817df98937fd). Extracted ROI responses and associated stimulus metadata, along with raw imaging data, can be found in a Dryad repository here: https://doi.org/10.5061/dryad. h44j0zpp8.

The following dataset was generated:

| Author(s) | Year | Dataset title | Dataset URL | Database and Identifier |
|---|---|---|---|---|
| Turner MH | 2022 | Data from: Visual and motor signatures of locomotion dynamically shape a population code for feature detection in *Drosophila* | https://dx.doi.org/10.5061/dryad.h44j0zpp8 | Dryad Digital Repository, 10.5061/dryad.h44j0zpp8 |

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
