## [Editor Report]

This manuscript investigates how the fly visual system can encode specific features in the presence of self-generated motion. Using volumetric imaging, it explores the encoding of visual features in population activity in the *Drosophila* visual glomeruli – a set of visual "feature detectors". Through an elegant combination of neural imaging, visual stimulus manipulations, and behavioral analysis, it demonstrates that two different mechanisms, one based on motor signals and one based on visual input, serve to suppress local features during movements that would corrupt these features. The results of this study open up new directions to determine how motor and visual signals are integrated into visual processing at the level of neural circuits.

---

## [Decision Letter]

**Decision letter after peer review:**

Thank you for submitting your article "Visual and motor signatures of locomotion dynamically shape a population code for feature detection in *Drosophila*" for consideration by *eLife*. Your article has been reviewed by 3 peer reviewers, and the evaluation has been overseen by a Reviewing Editor and Claude Desplan as the Senior Editor. The following individuals involved in the review of your submission have agreed to reveal their identity: Terufumi Fujiwara (Reviewer #1); Cristopher M Niell (Reviewer #2).

Essential revisions:

The reviewers appreciate the quality of the work presented in this interesting manuscript. The combination of neural imaging, visual stimulus manipulations, and behavioral analysis elegantly demonstrates that two different mechanisms, one based on motor signals and the other based on visual input, serve to suppress local features during movements that would corrupt these features. In spite of the high quality of the work, the reviewers raised several technical concerns that should be addressed prior to the publication of the manuscript. It is very likely that most of these points can be resolved through the analysis of existing data and/or appropriate editing of the main text. The reviewers agree that the addition of new experimental data can be minimized.

1) You should rule out that the correlated gain modulation observed in Figure 4 (and subsequent) is not due to motion artifacts or other factors that might vary during imaging. This control could be achieved by showing/analyzing red channel traces that you might already have. Alternatively, you could add a few caveats in the discussion about whether other factors might influence correlations across the population of VPNs.

2) In Figure 5, could the walking behavior be decomposed into forward and angular velocity components? This would strengthen the association between visual signals and specific behaviors. Through this analysis, it would be important to clarify the scope of "self-motion" by defining whether the visual inputs associated with rotations and forward movements are processed in the same way. Could the angular velocity range be computed during inter-saccade intervals of free-moving behavior to estimate the corresponding visual responses?

3) We encourage you to split the data in Figure 3D based on walking versus stationary states to demonstrate that the VPNs projecting to LC18 show the modulation seen in Figure 5C. This result would mitigate the possibility that the modulation by self-motion results from other inputs into the glomeruli that weren't completely eliminated by the genetic manipulations.

4) If possible, please complement the data presented in Figure 6 with a comparison of the activity observed upon rotational motion and stationary gratings.

5) Please motivate the idea that stimulus identity is encoded at the level of population activity and that positive correlations enhance stimulus decoding. The enhancement in stimulus decoding appears counter-intuitive. Related to this point, it would be helpful to improve the representation of the trial-to-trial correlations in a stimulus-dependent manner.

*Reviewer #1 (Recommendations for the authors):*

Figure 4

Even though the analysis looks quite reasonable, I have difficulty understanding how exactly the trial-to-trial activity correlation among glomeruli improves decoding visual feature identity. If the trial-to-trial correlation is totally random across identical visual stimulations, it will not provide extra information on visual feature identity. Therefore, the trial-to-trial correlation needs to be organized such that the shared activity (amplitude) is somehow specific to each visual feature stimulus. For example, the shared activity amplitude is always around 0.8 for looming and always 0.3 for single stripe, etc. Then, isn't such consistent activity already reflected in the total activity at each glomerulus? Alternatively, one possibility I could imagine is like following:

The activity of glomerulus A to looming: total activity = 1 (glomerulus-specific activity=0.2 + shared activity=0.8).

The activity of glomerulus A to single stripe: total activity = 1 (glomerulus-specific activity=0.7 + shared activity=0.3).

In this case, we cannot decode if the visual stimulus was looming or a single stripe from the total activity of glomerulus A, but we can decode if the total activity is divided into glomerulus-specific + shared activities. Is this the correct direction to interpret the result? Anyway, I wonder if the authors could provide a bit more intuitive explanation of how the shared activity contributes to the decoding.

Figure 8

Authors elegantly demonstrated that responses to local visual features are largely suppressed during visual and body saccades. On the other hand, it is not clear yet if the responses are not disturbed by suppression during inter saccade intervals or when the fly wants to process it. I wonder if the authors could estimate the angular velocity range during inter saccade interval from free moving behavior and estimate how many visual responses can be maintained in that range.

*Reviewer #2 (Recommendations for the authors):*

Overall, the study was well-designed and the data was presented clearly.

1. The authors make a compelling argument that they have restricted the glomerular signals to the PN terminals, and Fig 3 verifies that they match up in terms of mean response. However, it seems possible that some of the modulations by self-motion could represent either pre-synaptic modulation, or other inputs into the glomeruli that weren't completely eliminated with their genetic approach. Splitting the data in Fig 3D based on walking vs stationary and demonstrating that the VPNs projecting to LC18 also show the modulation seen in 5C would be a good way to confirm this.

2. The data in Fig 6A-G compellingly demonstrates that low SF stimuli suppress the response, but it's not clear that it is the rotational motion that is important since there is no comparison to stationary gratings. If that data is available (as it is for Fig 7) it would be very helpful, otherwise, it might be best to clarify that this data supports low SF stimuli suppressing, and the rotational effect is only shown later. On a related point, it is a little surprising that the coherent dots suppress the response since I would expect these to be more like the high SF / whitened stimuli of Fig 7.

3. The fact that visual stimuli can be decoded from the population in the presence of modulation by movement signals is quite similar to the findings of Stringer et al 2021 and Rumyantsev et al 2020, so it might be worth noting these.

*Reviewer #3 (Recommendations for the authors):*

1) My first concern is about whether any of the correlated gain modulations the authors observe could be due to motion artifacts or other factors that might vary during imaging but might not reflect actual neural signal intensity variations. This is a particular concern in Figure 4 where such shared variability is first introduced. Ideally, the authors would show imaging of the red channel from the same trials showing that this is not modulated by the shared gain factor. At the very least the authors should mention possible confounds that could give rise to this variability and discuss measures taken to rule these out.

2) Although the manuscript is framed in terms of "self-motion," most of the analysis and experiments focus on fast rotational motion evoked by body saccades. For example, the analysis in Figure 1 deals only with rotational motion, as do the visual suppression experiments in Figures 6 and 7. However, Figure 5 shows suppression driven by walking (not necessarily turning) and it is not clear from the figure if these represent rotations or forward movements. Therefore, it is not clear if the two inputs discussed are "working together" as suggested in the Discussion, or cover different types of input (during forward motion versus turns).

Although this does not detract from the interest of the work, it is confusing, as self-motion also includes large translational components which are not discussed (much) here and as saccadic suppression of visual signals has been discussed elsewhere. The authors should clarify in the Abstract and Introduction that their focus will be on rotational motion related to body saccades, and should address the differences between these types of motion in the Discussion.

3) The authors perform a decoding analysis of stimulus identity to argue that stimulus identity is encoded at the level of population activity and that positive correlations enhance stimulus decoding. This seems strange to me because classical studies of correlations in visual encoding (e.g. Shadlen Newsome) emphasized the way that correlated variability *reduces* the encoding capacity of a network. The emphasis on encoding stimulus identity within the particular stimulus set presented also seemed strange to me because it is not clear that the fly needs to discriminate between each of these stimuli in order to make appropriate behavioral responses. For example, flies are known to respond differently to vertical stripes versus short spots, however, it is not clear if they care about the difference between spots of slightly different sizes, or between spots moving on a gray versus grating background. Presumably, psychophysics experiments combined with connectomics can help determine which combinations of glomerular responses are actually used by the fly to shape its behavior. At any rate, I find that the conclusions about how VPNs encode visual features (e.g. Discussion line 439) rest on an assumption about what the fly is trying to do with these stimuli that may not be accurate.

---

## [Author Response]

Essential revisions:1) You should rule out that the correlated gain modulation observed in Figure 4 (and subsequent) is not due to motion artifacts or other factors that might vary during imaging. This control could be achieved by showing/analyzing red channel traces that you might already have. Alternatively, you could add a few caveats in the discussion about whether other factors might influence correlations across the population of VPNs.

Thank you for this suggestion. We have added example single trial red channel (myr::tdTomato) traces to Figure 4A, underneath the corresponding example syt1GCaMP6f traces. These structural signals show very little modulation from trial to trial relative to visually driven responses in GCaMP. We have also added a new Figure 4 —figure supplement 2 showing the trial covariance matrix for both red and green channel signals, showing much lower covariances in red channel signals compared to green channel signals, and a qualitatively different covariance structure. We address this concern and the results of these control analyses in the main text, beginning at line 231. To specifically look for an impact of animal movement on motion artifacts, we examined the correlation between red channel traces and animal walking behavior, and found no significant correlation for any glomerulus (see Author response image 1).

**Author response image 1. sa2fig1:** 

2) In Figure 5, could the walking behavior be decomposed into forward and angular velocity components? This would strengthen the association between visual signals and specific behaviors. Through this analysis, it would be important to clarify the scope of "self-motion" by defining whether the visual inputs associated with rotations and forward movements are processed in the same way.

This is an excellent question, and it has prompted us to substantially expand our analysis of free walking behavior. We have moved these data to a main figure (new Figure 8) to give them the focus they deserve. During free walking behavior, saccadic turns nearly always occur while the animal is also running forward, and forward velocities during a saccade are often no different than during an intersaccadic interval (Figure 8E). Because of this, periods of high rotational velocity cannot be isolated from forward velocity in freely moving animals. On the other hand, free walking flies do run forward without much rotational velocity, creating straight runs, but such runs are generally not seen in our fictive walking on the ball data, where movement bouts are interspersed with stationary periods. In our experience, these periods of “rest” are relatively common when flies are tethered. Because of this, we cannot decompose fictive rotational from translational velocity components in any way that allows us to associate response gain changes with one or the other component in isolation. What the ball locomotion measurements do provide is a measure of when the fly is standing still and when it is engaged in a movement bout, which consists of both rotational and forward translational components. We have updated the text to reflect the fact that in Figure 5 we cannot disambiguate between these two specific types of movement (line 297), and we later use free walking statistics in the new Figure 8 to better motivate the focus on the rotational components of saccades. We also modified the discussion (line 595) to note the inability to disambiguate forward from rotational velocity using fictive ball walking data, as well as the idea that these locomotor components are not fully separable even under natural walking conditions.

Could the angular velocity range be computed during inter-saccade intervals of free-moving behavior to estimate the corresponding visual responses?

This is a great question, and one that we have explored extensively in response. We will address it first as it relates to the visual gain suppression, and then as it relates to the motor-related suppression. As noted above, because of differences in walking behavior between free walking and fictive ball walking it is difficult to make a direct comparison, but two pieces of evidence shed light on how much suppression would be expected to be associated with intersaccadic walking. First, our new Figure 8 now shows forward and rotational velocities conditioned on saccadic vs. intersaccadic interval. During intersaccadic periods, rotational velocities are typically small (<40 deg/sec), but they do occasionally extend into quite fast rotations (long tail of gray distribution in Figure 8E). Comparing this to the image-driven suppression we measured in Figure 7, it indicates that some glomeruli (LC11, 21, and 18) could experience visual suppression during intersaccadic intervals, whereas other glomeruli (LC6 and 26) would be relatively unaffected by the typical intersaccadic run. We now note this interesting difference in the discussion (line 584).

To address the motor-related gain control component of this question, we added a new Figure 5 —figure supplement 1 showing the relationship between population response gain and walking amplitude, which shows minimal gain suppression up until ~10 deg/sec walking amplitude, and strong gain suppression being reached by ~40 deg/sec. Again, because of the differences between free walking and fictive ball walking it’s difficult to make absolute comparisons of these numbers to rotational velocities seen during free walking, but it does

3) We encourage you to split the data in Figure 3D based on walking versus stationary states to demonstrate that the VPNs projecting to LC18 show the modulation seen in Figure 5C. This result would mitigate the possibility that the modulation by self-motion results from other inputs into the glomeruli that weren't completely eliminated by the genetic manipulations.

While we don’t have behavioral tracking data for an LC18-specific driver line, we do have such data using a split-Gal4 driver line for LC11, which also showed behavioral modulation in our pan-glomerulus imaging experiments. We have added these data to the new Figure 5 —figure supplement 2, and address this important control in the main text (line 316). This genetically targeted approach recapitulated what we saw using the population imaging approach, giving us confidence that the behavioral modulation we saw came from VPNs, not some other, unidentified, ChAT-positive cell types that have neurites in the glomerulus.

4) If possible, please complement the data presented in Figure 6 with a comparison of the activity observed upon rotational motion and stationary gratings.

We do not have data for probe responses on top of stationary gratings, but as reviewer 2 notes, the distinction between rotating and stationary backgrounds is made in Figure 7 using natural images. We have updated the text to reflect the fact that the gratings experiments alone are not evidence that the surround is selective for rotational motion (lines 368, 358). We also clarify how the experiments using natural images in Figure 7 and the coherent dots stimulus was designed to test for selectivity for coherent, rotational motion (lines 372, 396).

5) Please motivate the idea that stimulus identity is encoded at the level of population activity and that positive correlations enhance stimulus decoding. The enhancement in stimulus decoding appears counter-intuitive. Related to this point, it would be helpful to improve the representation of the trial-to-trial correlations in a stimulus-dependent manner.

We apologize for the confusion about the decoding effect of positive correlations. The original manuscript did not do a good job explaining what is known about the effect of positive correlations in heterogeneously tuned populations and how this is related to what we see in VPNs. We have modified the text to highlight two key aspects of how this works: First, we have better highlighted that correlations improve decoding ability given a set amount of total variance. The overall variability itself is not helpful, only its correlation structure relative to a null model with no correlations (see paragraph starting at line 273). Second, previous work (e.g. that of Zohary, Shadlen and Newsome) that shows a deleterious effect of positive noise correlations is based on a homogeneously tuned population of neurons that use averaging across many cells to estimate the stimulus. In the case of heterogeneously tuned populations, like VPNs, correlations can “shape” noise along directions in population response space such that it does not interfere with estimates of the stimulus. This relies on a decoding strategy that is different from arithmetic averaging, for example a decoder that can compare differences or relative activations of differently tuned populations (see updated text at line 539). We have also added some more references to theoretical results in the literature on this topic, as our decoding results are in line with this past work and we think what we are seeing is the result of these previously-described impacts of noise correlations.

Related to this issue, and in response to feedback from reviewer 3, we have also added a new Figure 4 —figure supplement 3 showing decoding model performance on a discrimination task using only four visual stimulus classes that are known from previous behavioral work to elicit distinct visual behaviors (grating, spot, loom and vertical bar). That is, these are stimulus classes that the fly certainly can distinguish. Similar to the main figure, this new analysis also shows that most glomerulus groups encode information about most stimuli, and that for at least some stimuli, the population contains significantly more information than a single group alone. This is in line with our conclusion that VPNs encode most visual features in a distributed fashion, rather than single glomeruli being responsible for encoding single visual features.

Reviewer #1 (Recommendations for the authors):Figure 4Even though the analysis looks quite reasonable, I have difficulty understanding how exactly the trial-to-trial activity correlation among glomeruli improves decoding visual feature identity. If the trial-to-trial correlation is totally random across identical visual stimulations, it will not provide extra information on visual feature identity. Therefore, the trial-to-trial correlation needs to be organized such that the shared activity (amplitude) is somehow specific to each visual feature stimulus. For example, the shared activity amplitude is always around 0.8 for looming and always 0.3 for single stripe, etc. Then, isn't such consistent activity already reflected in the total activity at each glomerulus? Alternatively, one possibility I could imagine is like following:The activity of glomerulus A to looming: total activity = 1 (glomerulus-specific activity=0.2 + shared activity=0.8).The activity of glomerulus A to single stripe: total activity = 1 (glomerulus-specific activity=0.7 + shared activity=0.3).In this case, we cannot decode if the visual stimulus was looming or a single stripe from the total activity of glomerulus A, but we can decode if the total activity is divided into glomerulus-specific + shared activities. Is this the correct direction to interpret the result? Anyway, I wonder if the authors could provide a bit more intuitive explanation of how the shared activity contributes to the decoding.

Thanks for raising this point of confusion, which is the result of our poor explanation in the original manuscript. We have updated the text to better explain how positively correlated response variability can improve stimulus decoding relative to uncorrelated variability. See response to essential revision (5) above and updated text around lines 273 and 539.

Figure 8Authors elegantly demonstrated that responses to local visual features are largely suppressed during visual and body saccades. On the other hand, it is not clear yet if the responses are not disturbed by suppression during inter saccade intervals or when the fly wants to process it. I wonder if the authors could estimate the angular velocity range during inter saccade interval from free moving behavior and estimate how many visual responses can be maintained in that range.

Thanks for this suggestion. See response to major revision (2) above. We now address this point in the updated discussion (line 584) and with a new Figure 5 —figure supplement 1.

Reviewer #2 (Recommendations for the authors):Overall, the study was well-designed and the data was presented clearly.1. The authors make a compelling argument that they have restricted the glomerular signals to the PN terminals, and Fig 3 verifies that they match up in terms of mean response. However, it seems possible that some of the modulations by self-motion could represent either pre-synaptic modulation, or other inputs into the glomeruli that weren't completely eliminated with their genetic approach. Splitting the data in Fig 3D based on walking vs stationary and demonstrating that the VPNs projecting to LC18 also show the modulation seen in 5C would be a good way to confirm this.

This is a great suggestion and an important control. We did not collect behavioral tracking data during the split-Gal4 experiments in Fig. 3, but we do have behavioral tracking data using an LC11 split-Gal4 driver line, which shows a very similar behavioral modulation as we saw in the pan-glomerulus imaging approach. As noted above, we have added these new data to the new Figure 5 – supplement 2. As LC11 and LC18 both show strong behavioral modulation, we hope that these data address the main thrust of this concern.

2. The data in Fig 6A-G compellingly demonstrates that low SF stimuli suppress the response, but it's not clear that it is the rotational motion that is important since there is no comparison to stationary gratings. If that data is available (as it is for Fig 7) it would be very helpful, otherwise, it might be best to clarify that this data supports low SF stimuli suppressing, and the rotational effect is only shown later. On a related point, it is a little surprising that the coherent dots suppress the response since I would expect these to be more like the high SF / whitened stimuli of Fig 7.

Thank you - we were not clear on this in the original text. We have updated the text to clarify that the gratings experiments do not demonstrate a selectivity for rotational motion (line 368), they only characterize the spatial frequency and speed tuning of the surround (line 358). As you say, this point is made using coherent dots stimuli in Fig. 6 and natural images in Fig. 7 (updated text around line 390).

With regards to the suppression by coherent dots, it’s hard to make a direct comparison to gratings or filtered natural images, but it is important to note that the high spatial frequency gratings that fail to recruit strong surround suppression are composed of bars that are 2.5-5 degrees, whereas the individual dots that make up the coherent dots stimulus are close to 15 degrees. This is consistent with what we would expect to drive elementary motion detecting neurons T4/T5 given the spatial frequency tuning of those cells: 2.5-5 degree bars elicit weak T4/T5 responses, whereas a 15 degree spot will nearly fill the center of a T4/T5 receptive field. We have updated the text introducing this experiment (lines 372 & 378) to clarify the logic behind the experiment and more explicitly lay out that this stimulus was designed to recruit different subpopulations of T4/T5, and that we have some idea of what stimuli should drive T4/T5 from previous work.

3. The fact that visual stimuli can be decoded from the population in the presence of modulation by movement signals is quite similar to the findings of Stringer et al 2021 and Rumyantsev et al 2020, so it might be worth noting these.

Thanks - we have added this to the discussion.

Reviewer #3 (Recommendations for the authors):1) My first concern is about whether any of the correlated gain modulations the authors observe could be due to motion artifacts or other factors that might vary during imaging but might not reflect actual neural signal intensity variations. This is a particular concern in Figure 4 where such shared variability is first introduced. Ideally, the authors would show imaging of the red channel from the same trials showing that this is not modulated by the shared gain factor. At the very least the authors should mention possible confounds that could give rise to this variability and discuss measures taken to rule these out.

Thank you for raising this important concern. We have added traces from the red channel for the same trials shown in Figure 4A and added Figure 4 —figure supplement 2 which shows the covariance matrices for the green and red channels. The covariance in the red signal is across the board weaker than what is seen in the green channel, and its structure across the population looks different than the shared gain revealed by covariance analysis of GCaMP signals, indicating that the apparent shared gain is the result of visually driven responses, not motion correction errors or other imaging factors.

2) Although the manuscript is framed in terms of "self-motion," most of the analysis and experiments focus on fast rotational motion evoked by body saccades. For example, the analysis in Figure 1 deals only with rotational motion, as do the visual suppression experiments in Figures 6 and 7. However, Figure 5 shows suppression driven by walking (not necessarily turning) and it is not clear from the figure if these represent rotations or forward movements. Therefore, it is not clear if the two inputs discussed are "working together" as suggested in the Discussion, or cover different types of input (during forward motion versus turns).Although this does not detract from the interest of the work, it is confusing, as self-motion also includes large translational components which are not discussed (much) here and as saccadic suppression of visual signals has been discussed elsewhere. The authors should clarify in the Abstract and Introduction that their focus will be on rotational motion related to body saccades, and should address the differences between these types of motion in the Discussion.

Thank you for this feedback. We have clarified in the abstract, introduction (line 100), results (lines 115, 355, 372, 390), figure legend titles and subsection headings that our focus is on rotational visual motion specifically. We also discuss this choice and the differences between rotational and translational motion in the discussion (line 590). Finally, we have also modified the text to reflect the fact that forward and rotational components of self motion cannot be completely dissociated in natural walking or in our fictive walking data (see response to major revision (2) above).

3) The authors perform a decoding analysis of stimulus identity to argue that stimulus identity is encoded at the level of population activity and that positive correlations enhance stimulus decoding. This seems strange to me because classical studies of correlations in visual encoding (e.g. Shadlen Newsome) emphasized the way that correlated variability reduces the encoding capacity of a network. The emphasis on encoding stimulus identity within the particular stimulus set presented also seemed strange to me because it is not clear that the fly needs to discriminate between each of these stimuli in order to make appropriate behavioral responses. For example, flies are known to respond differently to vertical stripes versus short spots, however, it is not clear if they care about the difference between spots of slightly different sizes, or between spots moving on a gray versus grating background. Presumably, psychophysics experiments combined with connectomics can help determine which combinations of glomerular responses are actually used by the fly to shape its behavior. At any rate, I find that the conclusions about how VPNs encode visual features (e.g. Discussion line 439) rest on an assumption about what the fly is trying to do with these stimuli that may not be accurate.

We apologize for poorly explaining the effect of positive noise correlations in heterogeneous populations. See response to essential revision (5) above. Of particular note, we highlight that the information limiting correlations explored by Zohary, Shadlen and Newsom are in the case of a homogeneously tuned population, and as you say positive correlations in such a population limit the accuracy with which the population average response can be estimated, thereby interfering with stimulus decoding. Our decoding model, however, can compare activations across differently tuned populations, thereby taking advantage of the fact that in a heterogeneously tuned population, correlations can shape noise in response space so as to not limit the information available about the stimulus. We have improved the explanation for this result and its relation to currently existing theoretical work on this topic (see lines 273 and 539).

Regarding the second point, we have added a new Figure 4 —figure supplement 3 showing decoding for a reduced subset of stimuli that are known to be discriminable from past behavioral work. We agree that we should have acknowledged how assessing the information available within a neural population is only part of the story, and that understanding how this information is used to guide behavior is critical. We have updated the text to reflect this fact (line 524, discussion).